# Dbf4-dependent kinase finetunes Ino80 function at chromosome replication origins

Priyanka Bansal[1,10], Shibojyoti Lahiri[1,10], Chandni Natalia Kumar [1], Jessica Furtmeier [1], Lorenz Spechtenhauser[1], Lorenzo Galanti[2,6], Juan de Dios Barba Tena [2], Erika Chacin[1,7], Garp Linder[3], María Ángeles Ortíz-Bazán[4], Marisa Müller[1], Petra Vizjak [5,8], Tobias Straub [1], Felix Mueller-Planitz [5], Johannes Stigler [3], Andrés Aguilera [4], Belen Gómez-González [4], Boris Pfander [2], Philipp Korber [1], Axel Imhof [1]✉ & Christoph F. Kurat [1,9]✉

The highly conserved Dbf4-Dependent Kinase (DDK) plays a pivotal role during S phase. It phosphorylates the replicative helicase (minichromosome maintenance, MCM complex), which leads to the initiation of replication. However, few other targets, besides the MCM complex, are known, leaving DDK an understudied kinase. Here, we determine the nuclear DDK-dependent phosphoproteome by a two-pronged mass spectrometry approach. Among ~ 400 DDK-dependent phosphorylation targets, we find the Arp8 subunit of the INO80 chromatin remodeling complex. Arp8 phosphorylation stabilises INO80's complex integrity, finetunes its nucleosome spacing at replication origins, stimulates replication and improves the replication stress response. Taken together, we report the regulation of a chromatin remodeler with nucleosome-spacing activity by the cell-cycle machinery. DDK not only regulates the core replication machinery but also regulates a factor that generates replication-conducive chromatin architecture at replication origins.

The replication of nuclear genomic DNA represents one of the most fundamental processes in cells. Any impairment of nuclear DNA replication may lead to hyperproliferation and compromise genome integrity, both of which are hallmarks of cancerogenesis. Therefore, it is of the utmost importance that replication is tightly regulated and that repair mechanisms respond quickly and effectively to replication stress. Both replication and repair have to navigate chromatin, the packaged form of genomic DNA in the nucleus. In particular, genomic DNA is wrapped around histone octamer proteins forming nucleosomes, the basic units of chromatin. Nucleosomes are relatively stable, which provides a means for the safe storage and packaging of free DNA[1]. However, this is a challenge for DNA-templated processes such as replication, repair and transcription. Accordingly, the replication machinery requires the assistance of factors that deal with chromatin, like the histone chaperone complex FACT/Nhp6a, during replication through a chromatin template in a biochemically reconstituted

[1]Biomedical Center Munich (BMC), Division of Molecular Biology, Faculty of Medicine, Ludwig-Maximilians-Universität München, Munich, Martinsried, Germany. [2]Cell Biology, Dortmund Life Science Center (DOLCE), TU Dortmund University, Department of Chemistry and Chemical Biology, Dortmund, Germany. [3]Gene Center and Department of Biochemistry, Ludwig-Maximilians-Universität München, Munich, Martinsried, Germany. [4]Centro Andaluz de Biología Molecular y Medicina Regenerativa-CABIMER, Universidad de Sevilla-CSIC, Seville, Spain. [5]Institute of Physiological Chemistry, Faculty of Medicine Carl Gustav Carus, Technische Universität Dresden, Dresden, Germany. [6]Present address: DSB Repair Metabolism Laboratory, The Francis Crick Institute, London, UK. [7]Present address: Bayer AG, Pharmaceuticals, Research and Development, Genomic Medicine, Aprather Weg 18a, Wuppertal, Aprath, Germany. [8]Present address: Early-Stage Bioprocess Development, Boehringer Ingelheim Pharma GmbH & Co. KG, Biberach an der Riss, Germany. [9]Present address: Chair of Genetics, University of Bayreuth, Bayreuth, Germany. [10]These authors contributed equally: Priyanka Bansal, Shibojyoti Lahiri. ✉e-mail: Imhof@lmu.de; Christoph.kurat@bmc.med.lmu.de

system[2,3]. ATP-dependent remodelling enzymes with nucleosome spacing activity, such as INO80 or ISW1a, are essential for in vivo-like replication rates, indicating that regular nucleosome spacing plays a pivotal role[2]. Indeed, recent work demonstrated that optimal nucleosome spacing around replication origins, facilitated by the ATP-dependent remodelers INO80, ISW1a, ISW2 and Chd1, is crucial for efficient replication initiation[4].

DNA replication is a two-step process[5,6]. In the G1 phase of the cell cycle, the Origin Recognition Complex (ORC) loads the inactive replicative DNA helicase, the MCM complex, onto origins of replication with the assistance of other loading factors and ATP. In budding yeast, origins are short DNA sequence motifs[7-10] within a nucleosome-free region (NFR) flanked by the aforementioned arrays of regularly spaced nucleosomes on both sites[11-13]. It is noteworthy that, in addition to its canonical role as the MCM loader, ORC plays a second essential role prior to replication initiation in the G1 phase[4]. ORC is a boundary element for the INO80, ISW1a, ISW2 and Chd1 remodelers that enables them to generate the phased nucleosome arrays around origins.

Subsequently, during S phase, the MCMs are activated and separate the DNA strands as a prerequisite for the initiation of replication (origin firing)[5,6]. The number of origins is proportional to the size of the genome. For instance, budding yeast employs approximately 300 core origins to replicate its 12 mega base genome[7-10], whereas in human cells, it is estimated that approximately 30,000 origins are required, which correlates with the roughly 100-fold larger human genome[14]. Origins do not fire simultaneously. Rather, they follow a temporal programme (replication timing) of early- versus late-firing origins[15-22]. Proper execution of this programme is crucial for the regulation of replication and for genome integrity. However, the underlying molecular mechanisms remain poorly understood.

In addition to replication factors, two prominent cell cycle kinases, Cyclin-Dependent Kinase (CDK) and Dbf4-Dependent Kinase (DDK), are responsible for the initiation of replication by activation of the MCM complex[5,6]. CDK is a well-studied kinase with many target proteins involved in a plethora of cellular processes[23-25]. The levels of kinase-activating proteins, called cyclins, oscillate in cyclin-specific patterns during all major cell cycle stages, ensuring that CDK activity is regulated throughout the cell cycle[26]. In contrast, the kinase subunit of yeast DDK, Cdc7, has only one activator, Dbf4, which accumulates in S phase, where it is essential for the initiation of replication[27-32]. In addition to DDK's pivotal role in S phase, Dbf4 levels remain high until the metaphase-to-anaphase transition, and DDK is involved in processes like meiotic recombination[33,34], chromosome segregation[35-38], DNA double-strand break repair[39,40] and blocking over-replication[41]. Unexpectedly, DDK was found to interact with ORC at early-replicating origins already in the G1 phase[42-44]. The biological significance of this interaction remains to be elucidated. In contrast to CDK, only a limited number of target proteins have been identified for DDK, which continues to make DDK a somewhat enigmatic kinase.

The objective of this study was to gain further insights into the biology of DDK by elucidating DDK-dependent phosphorylation in nuclear processes. To this end, we developed a two-pronged mass spectrometry-based screening pipeline by inhibiting DDK function in vivo via two independent methods, followed by the enrichment of nuclear fractions and the analysis of the nuclear phospho-proteomes. A total of approximately 400 DDK-dependent phosphorylation targets were identified, mainly in chromatin-associated proteins.

Among these DDK target sites, we identified two phosphorylation sites within an unstructured region of the Arp8 subunit of the INO80 chromatin remodelling complex. INO80 localises to replication origins in vivo[45-47], and we previously identified a critical INO80 function in establishing nucleosome arrays at replication origins[4]. This led us to wonder whether DDK-dependent Arp8 phosphorylation might regulate this INO80 function. Indeed, conversion of the two serine residues in Arp8 to alanine, where we identified DDK-dependent

phosphorylation, showed a pronounced structural rearrangement of important modules of INO80 and reduced ATP hydrolysis by INO80. In line with our hypothesis, DDK-mediated phosphorylation of Arp8 was essential for positioning nucleosomes correctly at replication origins, thereby stimulating replication in vitro. Together with the biochemical findings, in vivo data demonstrated that the loss of Arp8 phosphorylation resulted in a replication defect and a severe growth phenotype under replication stress conditions.

Collectively, our findings indicated that DDK not only regulates known factors of replication initiation, but also factors that regulate chromatin architecture at replication origins. This highlights that the origin function has to be viewed as an interplay of classical replication machinery as well as chromatin factors. To the best of our knowledge, our data provide evidence of the regulation of a classical nucleosome-spacing remodeler by cell-cycle machinery. The chromatin-related role of DDK is mediated by a mechanism that "finetunes" precise nucleosome positioning through the action of a remodeler with nucleosome spacing activity. We propose a model whereby this mechanism may contribute to the replication timing programme and the maintenance of genome integrity.

## Results
### The nuclear phosphoproteome of DDK targets
The objective of this study was to expand our knowledge about DDK-dependent phosphorylation of proteins using a global proteomics approach, i.e., comparison of phosphopeptide enrichment in the presence and absence of DDK function in vivo. A comparable approach was recently published, in which DDK-dependent phosphoproteomes were identified in M-phase-arrested budding yeast cells[39]. As our primary interest was in chromatin processes during replication, we focused on phosphoproteomes of nuclear fractions from S-phase cells.

We considered various methods for interfering with DDK function. DDK is essential in yeast, so we employed the temperature-sensitive allele cdc7-4[48] for ad hoc ablation of Cdc7 activity, the kinase subunit of DDK, upon shift from the permissive (25 °C) to the restrictive temperature (37 °C) (Fig. 1a). DDK-dependent phosphorylation will be reduced or even absent upon ablation of the kinase at the restrictive temperature. However, the temperature upshift will also induce the heat shock response, which may confound results.

Therefore, we considered an alternative DDK inactivation approach. The comparison of two orthogonal DDK-affected datasets helps to define a "high-confidence" DDK-dependent phosphoproteome. It is known that phosphorylation of the MCM complex by DDK is inhibited by activation of the DNA replication checkpoint. An active checkpoint results in the binding of DDK by the checkpoint kinase Rad53 (mammalian CHK1), which phosphorylates DDK and thereby inhibits DDK binding to the MCM complex[49-52]. This prevents MCM phosphorylation and subsequent origin firing under replication stress. As we were primarily interested in DDK targets that were phosphorylated under similar conditions as the MCM complex, we reasoned that triggering the DNA replication checkpoint should inhibit the phosphorylation of such DDK targets, too. Cells were treated with the replication drug hydroxyurea (HU), which decreases dNTP pools and thus causes replication stress. Of note, both the incubation of the cdc7-4 mutant[53] as well as treatment with HU led to cell cycle arrest, in contrast to control conditions at the permissive temperature or without HU, respectively. This may also cause side effects to bear in mind.

We conducted a phosphoproteome analysis of nuclear extracts (Fig. 1a) from cdc7-4 cells under permissive or restrictive conditions (Fig. 1b, left panel) and from cells that were either treated or not treated with HU (Fig. 1b, right panel). As anticipated, differential phosphorylation was observed for both DDK ablation approaches, consistent with the pivotal role of phosphorylation in regulating the cell cycle. Both DDK inhibition approaches as well as the phospho-proteomics pipeline were validated by the observed decrease of

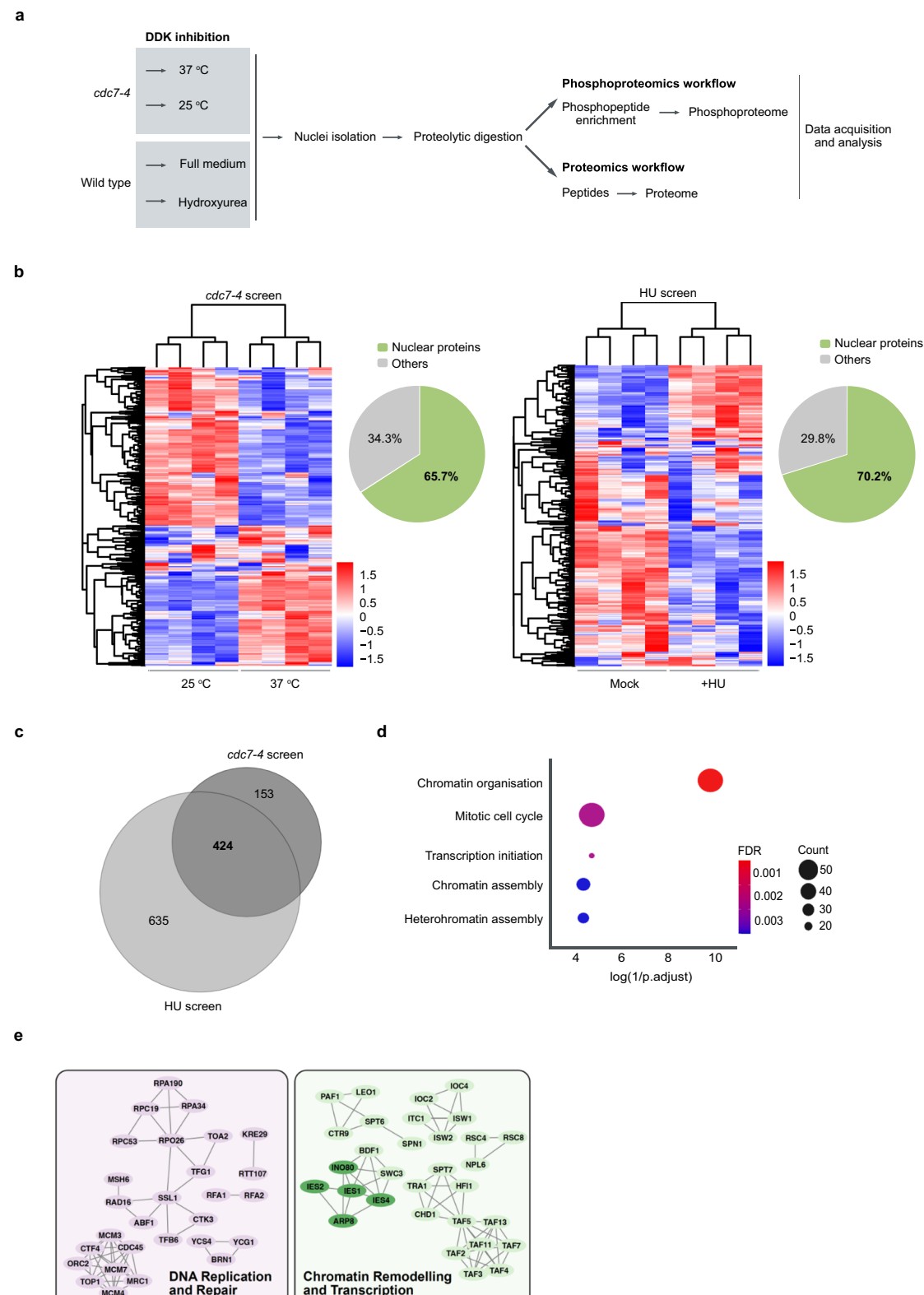

phosphorylation at the known DDK-dependent phospho-sites in the Mcm4 and Mcm6 subunits (Supplementary Fig. 1). DDK inhibition also affected other subunits of MCM (Supplementary Fig. 1). The biological consequences of this remain to be determined.

In order to reliably identify other DDK-dependent phosphorylation sites, we focused on the 424 targets that were enriched in both conditions with active DDK kinase (overlap in Venn diagram in Fig. 1c). A Gene Ontology (GO) analysis of these 424 DDK-dependent targets

revealed chromatin organisation as the most significant process regulated by DDK in vivo (Fig. 1d). A network analysis of all bona fide nuclear proteins that showed a significant increase in phosphorylation at the permissive temperature or without HU revealed several large complexes involved in DNA replication and transcription (Fig. 1e and Supplementary Fig. 2). As we recently demonstrated that the INO80 complex plays a pivotal role in establishing regular spacing of nucleosomes around replication origins[4], we were intrigued by

**Fig. 1 | The nuclear DDK-dependent phosphoproteome. a** Schematic representation of the proteomic workflow to determine the DDK-dependent phosphoproteome. **b** Heatmaps showing the clustering of top 500 variable phosphosites in the *cdc7-4* (left map) and Hydroxyurea (HU) screens (right map), respectively. Each column of the heatmap is a replicate of the corresponding condition (25 or 37 °C; Plus (HU) or minus HU (Mock). *Insets*: Pie charts showing nuclear enrichment of the phosphoproteins in the respective screens. The temperature screen shows an enrichment of 65.7%, whereas the HU screen shows 70.2% enrichment. *N* = 4 (for each condition). **c** Venn-diagram showing the overlap of parent proteins of all downregulated phosphosites in the 37 °C and HU fraction of the *cdc7-4* and HU screens. The comparative analysis was performed on all the down-regulated sites along with the exclusive sites. Sites were considered exclusive to a condition (e.g., permissive temperature or no HU treatment) when the perturbed condition (e.g., restrictive temperature or HU treatment) had the same sites detected in none or just 1 replicate. **d** GO term analysis using the corresponding proteins of the 424 overlapping phosphosites. Plots for the top 5 most significant biological processes that are over-represented in the overlapping phosphosite dataset are shown here. The plot was generated using the R package 'ggplot'. **e** Shown are all bona fide nuclear proteins that show DDK-dependent phosphorylation in both screens and have at least one high-confidence interactor. Grouping was done manually. Subunits of the INO80 complex are highlighted in dark green.

numerous INO80 components being phosphorylated when DDK was active (Fig. 1e). However, the experimental setup could not distinguish between sites that were directly phosphorylated by DDK versus sites that were phosphorylated by a downstream kinase or indirect consequences of the cell cycle arrest.

## The Arp8 subunit of INO80 is a bona fide DDK target

We observed that the purified INO80 complex and DDK interact physically, which provides evidence that the INO80 complex is a DDK target (Supplementary Fig. 3a). To validate which INO80 subunit was a direct DDK substrate, we then conducted an in vitro kinase assay with purified INO80 complex and DDK. In a pilot experiment, we used the assay to phosphorylate DDK's known target: the MCM complex. The Mcm4 and Mcm6 subunits, both of which are confirmed major DDK targets[32,49,54–56], were clearly phosphorylated by DDK in vitro (Supplementary Fig. 3b). The other subunits were not phosphorylated, demonstrating the specificity of our assay. INO80 contains 15 subunits and numerous serine and threonine residues. Only one subunit with approximately the size of the Arp8 subunit (Fig. 2a) was significantly phosphorylated by DDK in vitro (Fig. 2b, c and d). Serine 65 and 233 in Arp8 were phosphorylated by DDK in our phosphoproteome datasets (Figs. 1, 2a). We mutated these serine residues to alanine and purified the resulting complex (INO80-AA, Fig. 2a and b). These mutations did not modify the subunit stoichiometry of INO80-AA compared to INO80 (Fig. 2b). However, these mutations did strongly reduce the ability of DDK to phosphorylate Arp8 in the complex (Fig. 2c, d). This validated serine 65 and 233 in the Arp8 subunit of INO80 as direct targets of DDK.

## Arp8 phosphorylation is important for intramolecular integrity of INO80

Next, we investigated the consequences of a loss of Arp8 phosphorylation by DDK. INO80 is an approximately 1 MDa chromatin remodeler of 15 subunits that are organised into three structural modules: the "N-module" (Nhp10/Ies1/Ies3/Ies5), "A-module" (Arp4/Arp8/Act1(actin)/Taf14/Ies4) and "C-module" (Rvb1/Rvb2/Arp5/Ies2/Ies6) (Fig. 2e)[57–59]. The Ino80 subunit carries the motor ATPase, which has ATP-dependent DNA tracking activity and pumps extranucleosomal DNA into the nucleosome core particle as part of the sliding mechanism[57]. The Ino80 subunit also serves as a scaffold for the assembly of all three modules. The function of the N-module, which is least conserved, is least understood but involved in INO80's nucleosome spacing activity[60,61]. The A module, which contains Arp8, binds extranucleosomal DNA[58,62–67] and likely regulates the motor ATPase Ino80 in the context of DNA length sensing. The unstructured region at the N-terminus of Arp8, encompassing serine 65 and 233 (Fig. 2a), is important for this function[62] and involved in the nucleosome spacing activity of INO80 in vivo[68].

As no differences were observed in the stoichiometric composition of INO80 compared to INO80-AA (Fig. 2b), we hypothesised that Arp8 phosphorylation may affect the conformational organisation of the complex. However, this unstructured N-terminal part of Arp8 was

not amenable to resolution via cryo-electron microscopy[57]. Instead, we probed the subunit interaction topology of INO80 versus INO80-AA by cross-linking coupled to mass spectrometry, which offers a lower resolution compared to cryo-electron microscopy, but also informs on dynamic interactions of unstructured regions. We used a similar protocol as previously established for the INO80 complex[59] (Fig. 2f), but titrated anew the optimal concentration of the MS2-cleavable DSBU crosslinker for both complexes (Supplementary Fig. 4a).

In accordance with previous work[59], the three N-, A- and C-modules organised by the Ino80 scaffold were identified for INO80 (Fig. 2g). The N-module was crosslinked with the N-terminal part, the A-module with the middle part, and the C-module with the C-terminal part of Ino80. For the INO80-AA complex, the number of inter- and intracrosslinks was altered compared to INO80 (Fig. 2g). Most strikingly, the number of crosslinks between the N-module subunits and the N-terminal part of Ino80 was significantly reduced, and crosslinks between Ies1 and the rest of the N-module were virtually absent (Fig. 2g).

Cross-linking experiments are very sensitive to experimental conditions, including cross-linker concentrations and the quality of protein preparations. Therefore, we repeated the crosslinking experiments with an independent batch of purified INO80 complexes. Despite some variation in the crosslinking patterns, we reproduced a similar crosslinking pattern between the Ino80 scaffold and the N-, A- and C-modules (Supplementary Fig. 4b) and, most importantly, did reproduce the extreme change regarding the interactions between Ino80 and the N-module.

Collectively, a non-phosphorylated Arp8 subunit of INO80 correlated with a change in subunit topology, especially regarding the N-module.

## Arp8 phosphorylation stimulates INO80 ATPase and nucleosome sliding activity

As Arp8 phosphorylation affected the N-module and as the N-module has a role in nucleosome sliding by INO80, we tested how DDK-dependent phosphorylation of serines 65 and 233 affected ATP hydrolysis by the Ino80 motor ATPase and/or nucleosome sliding by INO80.

We employed an ATPase assay where ATP hydrolysis is coupled to the conversion of phosphoenolpyruvate to lactate. The concomitant turnover of NADH is followed photometrically and allows the calculation of ATP hydrolysis rates (Fig. 3a). The INO80-AA mutant complex showed a ~ 5-fold reduction in the ATP hydrolysis rates compared to the INO80 complex (Fig. 3b). ATP hydrolysis was strongly stimulated by DNA, as published[69].

Given this dependence on DNA, the reduction in ATP hydrolysis rate of the INO80-AA mutant complex could have resulted from decreased DNA binding. To address this possibility, we assessed DNA binding by both complexes by a single-molecule DNA curtain approach[70]. Both complexes were fluorescently tagged via site-specific ybbR labelling and subsequently injected into flow cells, which contained lambda DNA molecules tethered to a lipid bilayer and aligned at

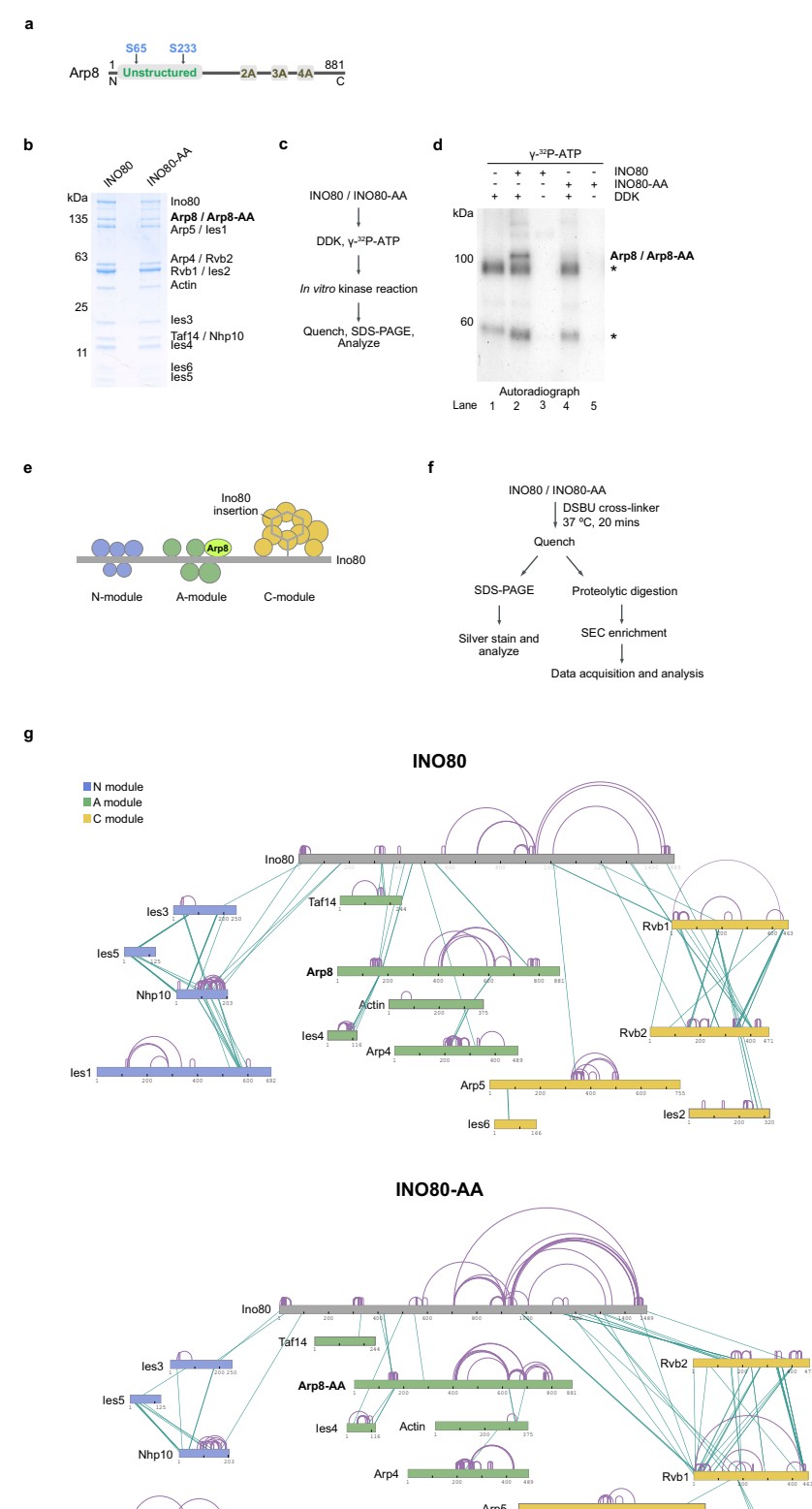

nano-fabricated barriers (Supplementary Fig. 5a). The number of complexes bound to the DNA was counted using fluorescence microscopy. There was no significant difference between the INO80 and INO80-AA complexes in their binding to lambda DNA (Supplementary Fig. 5b, c). Thus, the decrease of ATPase activity in the absence of DDK-dependent phosphorylation of the Arp8 subunit was not due to impaired DNA binding.

For remodelers such as SWI/SNF, it is known that their ATPase activity is about equally stimulated by DNA or nucleosomes[69]. In contrast, INO80's ATPase activity is more stimulated by nucleosomes than by DNA[58]. Accordingly, we also observed a stronger stimulation of ATPase activity with mononucleosomes compared to DNA for the wild-type INO80 complex (Fig. 3b). However, the ATPase activity of the INO80-AA mutant complex was less stimulated by mononucleosomes

**Fig. 2 | The Arp8 subunit of the INO80 complex is a bona-fide substrate of DDK.** **a** omain organisation of the Arp8 Subunit of the INO80 complex. Highlighted is the unstructured region and the two identified DDK-dependent phosphorylation sites, serine 65 and serine 233. 2A, 3 A and 4 A represent non-actin folds. **b** SDS-PAGE analysis of purified INO80 wild type and INO80-AA mutant complexes. $N = 5$ correspond to biological replicates. **c** Reaction scheme for the in vitro kinase reaction using purified DDK, INO80 and INO80-AA complexes. **d** Incorporation of $[^{32}P]$-γ-ATP into INO80 and INO80-AA by DDK was visualised by autoradiography after separation using SDS-PAGE. The presence of asterisks indicates the occurrence of auto-phosphorylation of DDK, as determined through a reaction with DDK alone (Lane 1). $N = 2$ correspond to independent biological replicates. **e** Schematic

depicting the organisational structure of the INO80 complex's subunits and submodules. The Arp8 subunit is highlighted. The Submodules "N" (NHP10 module), "A" (ARP8 module) and "C" (Core module) are colour coded in blue, green and yellow, respectively. **f** Outline of the cross-linking reactions using ureido-4,4´-dibutyric acid bis (hydroxysuccinimide) ester (DSBU) crosslinker with purified INO80 and INO80-AA complexes. **g** Crosslink network map of INO80 and INO80-AA complexes. Different subunits and modules of INO80 are colour-coded in accordance with (E). Inter cross-links are represented in green, and intra cross-links are represented in purple within the complex. $N = 2$ (see also supplementary data Fig. 3). Source data are provided as a Source Data file.

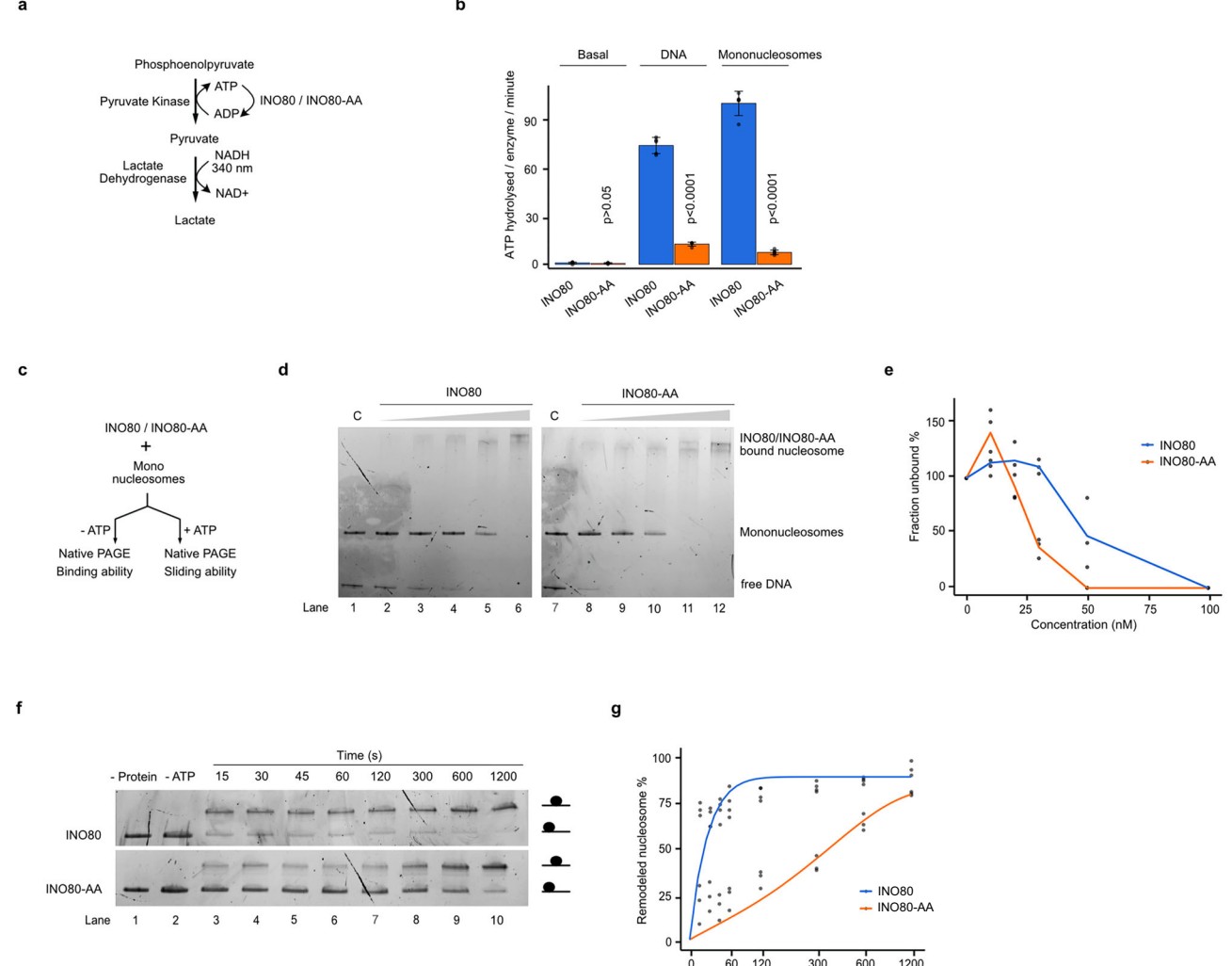

**Fig. 3 | ATPase and mononucleosome-mobilisation experiments of INO80 wild type and INO80-AA mutant complexes. a** Outline of the assay used to measure the ATP hydrolysis rates of the purified INO80 and INO80-AA complexes. **b** ATP hydrolysis rates of INO80 and INO80-AA complexes, both stimulated and unstimulated, in the presence of either naked DNA or mononucleosomes. The amount of ATP hydrolysed per enzyme per second was obtained by measuring the oxidation of NADH, which is used to regenerate ADP to ATP. The graph was plotted using standard mean and error, and $p$-values were obtained using two-tailed unpaired $t$ test calculations. $p$-value corresponds to $4.76 \times 10^{-9}$ and $3.81 \times 10^{-9}$ for DNA and mononucleosomes, respectively. $N = 5$ correspond to biological replicates. **c** Outline of the assay to measure the binding of the INO80 and INO80-AA complexes to mononucleosomes, and their ability to mobilise them. **d** The binding of the INO80 and INO80-AA complexes to mononucleosomes was assessed using an electrophoretic mobility shift assay (EMSA) coupled with native polyacrylamide gel

electrophoresis (PAGE). Binding was tested using purified complexes in the range 0–100 nM, as illustrated by the grey triangle at the top. Reaction C served as the control, containing only mononucleosomes. $N = 3$. **e** The unbound fractions (mononucleosomes) in (**b**) were quantified for all experiments and plotted against increasing remodeler concentrations. The results of the three experiments are presented as individual data points. **f** Analysis of the mobilisation of mononucleosomes by INO80 and INO80-AA complexes using native PAGE. $N = 3$. **g** The remodelling activity of the INO80 and INO80-AA complexes was assessed by quantifying the intensity of the bands corresponding to remodelled nucleosome species (i.e., bands with slower mobility in the gel). The fraction of remodelled nucleosomes was plotted over time, and the data were fitted to an exponential equation. The results of the three experiments are presented as individual data points. $N = 3$. Source data are provided as a Source Data file.

than by DNA and therefore even more - at least 9-fold - reduced compared to wild-type INO80 if incubated with mononucleosomes (Fig. 3b). Also, for the case of ATPase stimulation by mononucleosomes, we tested if the INO80-AA complex was affected in mononucleosome binding. For this, we could not employ the DNA curtain assay, but used electromobility shift assays (EMSA) with mononucleosomes assembled on 601 Widom nucleosome positioning sequences plus 80 bp linker DNA (0-N-80 mononucleosomes) and purified INO80 complexes without ATP (Fig. 3c). The appearance of slower migrating forms was indicative of INO80 complexes binding to mononucleosomes concomitant with the disappearance of the mononucleosome band (Fig. 3d). As the slower migrating high molecular forms were difficult to quantify, we measured only the intensities of the mononucleosomal bands. To our surprise, the INO80-AA mutant complex even showed increased binding affinity for nucleosomes compared to wild-type INO80 (Fig. 3d and e). This excludes that decreased ATP hydrolysis of the INO80-AA mutant complex was due to lower affinity to nucleosomes.

The decrease in ATP hydrolysis suggested that nucleosome sliding activity was compromised as well for the INO80-AA complex. We used a similar setup as for the EMSAs, but in the presence of ATP (Fig. 3c). The nucleosome spacing activity of INO80 corresponds to its ability to slide end-positioned mononucleosomes into the centre of the DNA fragment[71], which is detected by an upshift of the nucleosome band in native polyacrylamide gel electrophoresis (PAGE; Fig. 3f).

For the INO80 wild type complex, we observed nucleosome mobilisation to be both enzyme- and ATP-dependent and mostly completed within two minutes (Fig. 3f and g). In contrast, the INO80-AA mutant complex showed much slower nucleosome sliding kinetics with near complete remodelling after ~20 minutes (Fig. 3f, g). Such slower remodelling kinetics were consistent with the decreased ATP hydrolysis rates and demonstrated a strong effect of DDK-dependent phosphorylation of Arp8 on ATP hydrolysis and nucleosome mobilisation by INO80.

## Arp8 phosphorylation affects nucleosome spacing activity of INO80 and replication in vitro

Our experiments with mononucleosomal substrates demonstrated the impact of Arp8 phosphorylation on INO80 function. However, mononucleosomes are far from the physiological situation of nucleosome arrays. Therefore, we turned to our genome-wide chromatin reconstitution approach. The approach we also used to study the role of nucleosome arrays as generated at replication origins by remodelers like INO80, ISW1a, ISW2 and Chd1 in combination with ORC[4]. Now we tested how DDK-dependent phosphorylation of Arp8 affected this nucleosome array generation at replication origins. Purified histones were assembled by salt gradient dialysis (SGD) into nucleosomes on a library of ca. 400 plasmids with on average 13 kb inserts that comprised ca. 300 budding yeast replication origins as previously described[4] (Fig. 4a). The generation of nucleosome arrays at origins after incubation of this SGD chromatin with purified ORC and INO80 wild-type or INO80-AA mutant complexes in the presence of ATP was monitored by limited digestion with micrococcal nuclease (MNase), isolation of mononucleosomal DNA fragments and high-throughput sequencing (MNase-seq).

The MNase-seq data for the wild-type INO80 complex reproduced our previous results[4]. The combination of ORC and INO80 generated a nucleosome-free region (NFR) at ACS sites and bi-directional, phased arrays of regularly spaced nucleosomes (Fig. 4b). While the INO80-AA mutant complex generated an overall similar nucleosome organisation, there were still clear differences. The nucleosome positioning was less pronounced, i.e., the peak-to-trough ratios were dampened, and the linker lengths were on average longer (Fig. 4b).

Thus far, we based our conclusions about the role of DDK-dependent phosphorylation of serines 65 and 233 on the INO80-AA mutant, where these serines were replaced with alanine, barring phosphorylation. To address more directly whether DDK-dependent phosphorylation affected INO80 function, we dephosphorylated both the INO80 wild-type and INO80-AA mutant complexes by lambda phosphatase (Lambda PP). We then treated the complexes with DDK or not and re-purified the complexes (Supplementary Fig. 6a). Strikingly, the comparison of nucleosome positioning patterns generated by the INO80 wild-type complex treated or not with DDK (Supplementary Fig. 6b) re-capitulated the differences regarding the linker lengths between wild-type and mutant INO80-AA complexes, respectively (Fig. 4b). It should be noted that MNase-seq is very reliable with regard to nucleosome positions (peak positions), but less reliable and reproducible with regard to nucleosome occupancies (peak heights)[72,73]. Therefore, our conclusions are only based on comparisons of peak positions rather than relative peak heights.

In contrast, the nucleosome patterns generated by the INO80-AA mutant complex did not show much difference with or without DDK treatment.

Defects in nucleosome positioning when using the dephosphorylated INO80 complex may be due to altered nucleosome binding compared with the phosphorylated complex. We therefore tested the ability of the differentially phosphorylated INO80 complex to bind mononucleosomes using EMSAs. The results revealed similar nucleosome binding for both the unphosphorylated and phosphorylated INO80 complexes (Supplementary Fig. 6c, d). These findings support our hypothesis that DDK phosphorylation plays a direct role in INO80-mediated nucleosome organisation rather than regulating its affinity to nucleosomes.

Previously, we demonstrated that precise nucleosome positioning at replication origins is essential for replication initiation[4]. We therefore investigated whether defects in nucleosomal architecture at replication origins, as observed with the INO80-AA mutant complex (Fig. 4b), might cause replication problems. We tested this using our genome-scale, in vitro chromatin replication system[4] (Fig. 4c). Robust replication was observed when chromatin was assembled using the wild-type INO80 complex (Fig. 4d, lanes 1 and 2) but was significantly reduced when we used the INO80-AA mutant complex (Fig. 4d, lanes 3 and 4). Interestingly, the sizes of the leading strands were similar in both experiments, suggesting that replication initiation from a subpopulation of origins was defective (Fig. 4e).

Taken together, our biochemical assays provide direct evidence that DDK-dependent phosphorylation of the Arp8 subunit of INO80 is involved in its role in nucleosomal spacing. Furthermore, they suggested that this leads directly to defects in replication initiation, rather than elongation.

## DDK-dependent phosphorylation of INO80 is required for replication and during replication stress in vivo

As our biochemical experiments indicated an important role for DDK-dependent Arp8 phosphorylation in nucleosome sliding and generating the NFR-array organisation at replication origins, as well as for replication initiation, we proceeded to investigate the consequences of a loss of DDK-dependent phosphorylation of INO80 in vivo. We generated an *arp8-AA* allele and assayed nucleosome positioning around origins in vivo (Supplementary Fig. 7). The *arp8-AA* mutant showed weaker, but reproducible, dampening of array regularity compared to an isogenic *ARP8* strain than the respective differences in our in vitro assays (Fig. 4b). This may argue for some redundancy in vivo that we did not reconstitute in vitro. Despite the weak effects on nucleosome organisation around origins in vivo, we did observe a clear defect in replication for the *arp8-AA* mutant. The entry into S phase in synchronised cells after pheromone-mediated cell cycle arrest and release was delayed in the *arp8-AA* mutant compared to wild type (Fig. 5a). We observed an increased block in G1, where non-replicated DNA (1 C) remained at high levels for up to 50 minutes after release from α-

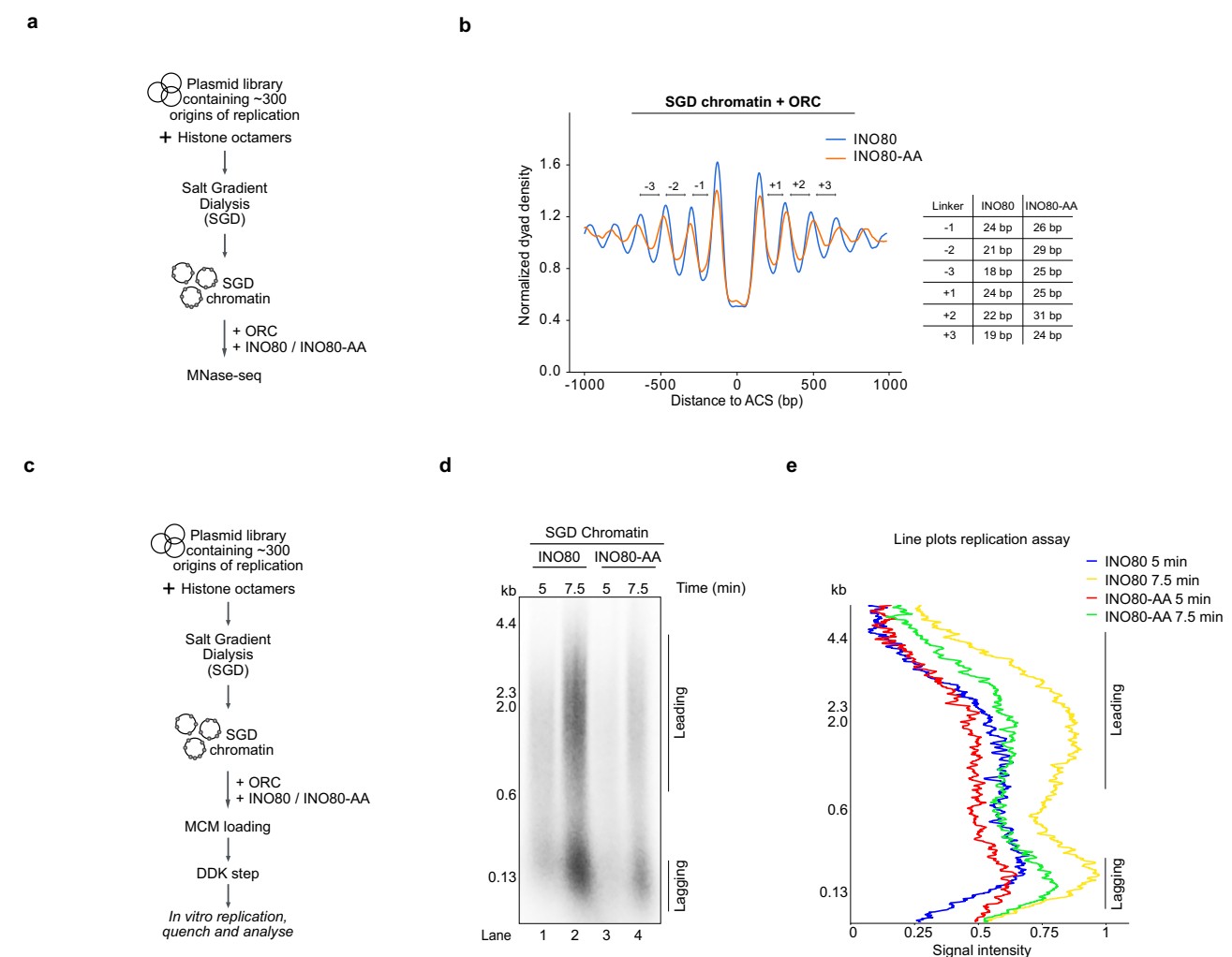

**Fig. 4 | DDK-dependent phosphorylation is important for nucleosomal spacing activity of the INO80 complex and influences replication in vitro. a** Outline of the genome-scale nucleosome reconstitution assay with purified INO80 and INO80-AA complexes to measure nucleosome spacing at origins of replication. **b** Composite plot illustrating representative in vitro MNase-seq data for SGD chromatin incubated with ORC plus either INO80 (blue) or INO80-AA (orange) complexes as described previously[4]. Approximately 300 replication origins are aligned to the ORC binding motif ACS (ARS (Autonomously Replicating Sequence) Consensus Sequence). The accompanying table presents the linker lengths of the first, second, and third linkers (DNA between nucleosomes) upstream and downstream of NFR (Nucleosome Free Region) generated by INO80 or INO80-AA complexes. $N = 2$. **c** Outline of the genome-scale chromatin replication assay to measure replication from chromatin assembled as in (**a** and **b**). **d** Genome-scale in vitro replication as in (**c**) of SGD chromatin assembled as in (**a** and **b**). Radiolabelled DNA products were separated in alkaline agarose gels and visualised using phosphorimaging. The leading and lagging strand replication products were visible because the assay did not include Okazaki fragment maturation factors. The figure shows a representative result of two independent replicates. $N = 2$. **e** Line plots of rchromatin eplication reactions as in (**d**). Line plots were generated using the FIJI version of ImageJ. Source data are provided as a Source Data file.

factor, only decreasing after 60 minutes. As cells did not accumulate in S phase, we could not distinguish between elongation and initiation using this assay. Nevertheless, the data are consistent with our in vitro replication results, which showed that initiation, rather than elongation, was affected when chromatin was assembled with the INO80-AA mutant complex (Fig. 4c, d, e).

Replication problems can lead to recombinogenic DNA damage. Indeed, we observed a twofold increase in levels of spontaneous homologous recombination for the *arp8-AA* mutant relative to wild-type, detected with a direct-repeat recombination assay (Fig. 5b). We next assayed the ability of the *arp8-AA* mutant to cope with different forms of replication stress. The *arp8-AA* mutant exhibited a minor growth phenotype compared to *ARP8* wild type on full media without any drug supplementation (Fig. 5c). A similar picture emerged when cells were grown on media with methyl methanesulfonate (MMS), which induces heat-sensitive DNA damage (Fig. 5c). In stark contrast, *arp8-AA* mutants were extremely sensitive to HU (Fig. 5c). We thus speculate that Arp8 phosphorylation is crucial for restarting

replication after replisomes have stalled due to limited NTP pools (i.e., in the presence of HU). We therefore analysed how *arp8-AA* responded to acute exposure to HU (Fig. 5d). Following a 2 h exposure to HU, *arp8-AA* cells were able to restart replication when released from the HU block, albeit with a significant delay compared to the wild type. Whether this restart defect was due to elongation defects (from already initiated forks) or initiation cannot be discriminated by these experiments.

As INO80 is also involved in nucleosome positioning at promoters[74], we asked whether DDK-dependent phosphorylation of INO80 affected transcription. In particular, we wondered if the replication defect phenotype in vivo may reflect indirect effects due to alterations in the expression of genes encoding cell cycle regulators (in particular those involved in G1 and S phase) or replication factors. We reasoned that, if transcription of genes involved in the G1 to S transition were affected in the *arp8-AA* mutant, this would only be visible at specific stages of the cell cycle. Consequently, we examined the transcriptome using RNA sequencing of synchronised cells released into

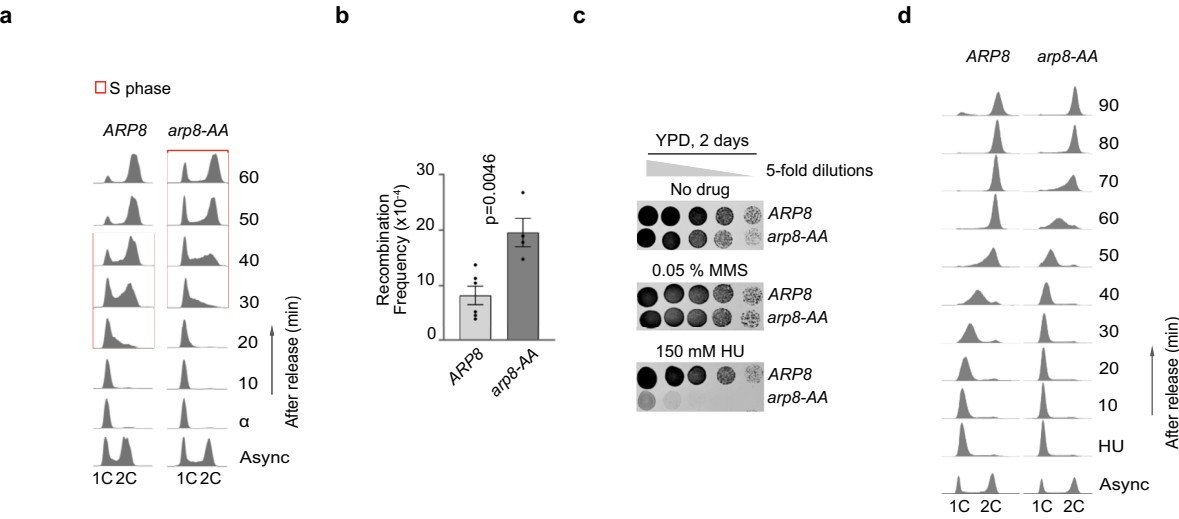

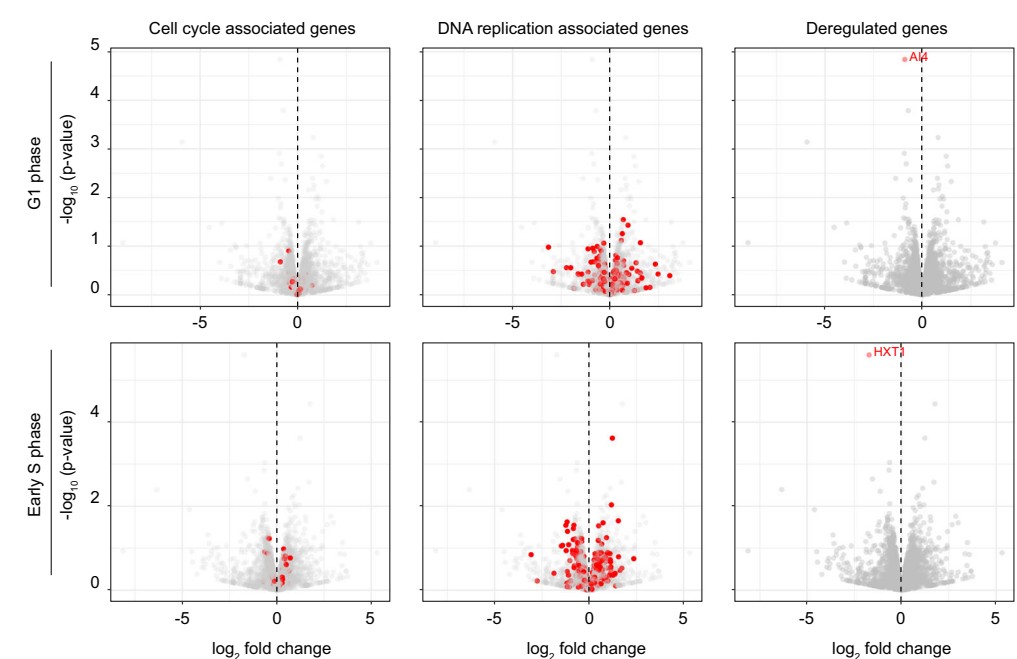

**f**

the G1 phase and the early S phase (Fig. 5e). No significant disparities in the transcription of cell cycle or replication genes were observed at either stage. The only two genes that were deregulated were metabolic genes, which were unlikely to cause a delay in the cell cycle. Thus, these analyses provide further support for our model, which proposes that replication defects in the *arp8-AA* mutant was caused by misaligned nucleosomes at replication origins rather than a failure in transcription

regulation. Analogous to our experiments with synchronous cells, we compared the transcriptomes of *ARP8* wild-type and *arp8-AA* strains grown asynchronously. Also here, no significant changes in the transcript levels of cell cycle or replication genes were observed (Supplementary Fig. 8). More generally, the group of deregulated genes was relatively small and mainly involved genes involved in metabolism (Supplementary Fig. 8). It is noteworthy that this group comprised

**Fig. 5 | DDK-dependent phosphorylation of INO80 is important for efficient replication in vivo. a** Flow cytometry profiles of *ARP8* versus *arp8-AA* cells before (async) and after synchronisation in G1 phase with α-factor (α) and release (time points after release are indicated to the right). 1 C and 2 C indicate the position of non-replicated and replicated DNA fluorescence signal, respectively. Time points with cells in S phase are marked with a red rectangle. *N* = 2. **b** Increase of recombination frequency in the *arp8-AA* mutant compared to *ARP8* WT. The graph is plotted with standard mean and error, and *p*-values was obtained by using a two-tailed unpaired *t* test calculation. *N* = 4 correspond to independent experiments. **c** Spot dilution growth assays (5-fold dilutions) with *ARP8* and *arp8-AA* strains. YPD: Yeast extract, Peptone, Dextrose (full medium); MMS: methyl methane sulphate and HU: hydroxyurea. *N* = 2. **d** Cell cycle analyses as in (**a**), comparing *ARP8* and *arp8-AA* cells released from an HU-induced S-phase block. 1 C and 2 C show the

positions of the non-replicated and replicated DNA fluorescence signals, respectively. *N* = 3. **e** Volcano plots showing differential expression of the *arp8-AA* mutant versus *ARP8* wild type. Cells were treated as in (**a**), and samples representative of the G1 and early S phases were taken. The red dots represent genes involved in the cell cycle (left panel), replication (middle panel), or significantly deregulated genes (right panel). The log$_2$ fold change shows the extent to which gene expression is either upregulated or downregulated in *arp8-AA*. *N* = 2, *p*-values from DESeq2 (Wald test; negative binomial GLM, adjusted for replicate. Lists of cell cycle and replication genes can be found in Supplementary Data 1. Source data are provided as a Source Data file. **f** Model depicting the role of DDK activity at early replication origins, highlighting its function in finetuning INO80 activity through phosphorylation. See text for details.

genes previously identified in a screen for transcriptionally deregulated genes under replication stress conditions[75] (Supplementary Fig. 8). This suggested again that the *arp8-AA* mutant experienced replication stress.

In conclusion, the combined results of our DDK-dependent phosphoproteome screens yielded approximately 400 phosphorylation targets with high confidence. Among these targets, we identified and validated serine 65 and 233 of the Arp8 subunit of the INO80 chromatin remodeler. Biochemical and in vivo assays demonstrated a role of DDK-dependent phosphorylation of Arp8 in regulating the nucleosome organising activity of INO80 at replication origins. Lack of DDK-dependent phosphorylation of Arp8 delayed replication and compromised survival under conditions of replication stress.

## Discussion

This study presents a comprehensive phosphoproteome of DDK targets in the budding yeast nucleus. A key feature of our approach was cross-validation by two independent methods to impair DDK activity (Fig. 1a). Each method had its own advantages and disadvantages. For instance, the elimination of DDK activity by a temperature-sensitive allele avoids side effects from chemicals such as kinase inhibitors. However, both the permissive and the restrictive temperature employed in the screen are not optimal for yeast. At 37 °C, heat shock-associated phosphorylation may lead to false positives, while at 25 °C, some phosphorylation that would occur at the optimal growth temperature of 30 °C may be underrepresented. Indeed, our observations indicated that serine 65 in Arp8 was particularly abundant at 30 °C. For the HU screen, the activation of checkpoint kinases addresses the temperature issue, but checkpoint kinases such as Rad53, in addition to influencing DDK binding to DDK targets, are likely to give rise to DDK-independent changes in phosphoproteomes. Consequently, a two-pronged approach as employed here is required to identify DDK targets with a high degree of confidence.

Of all identified DDK-targets, we focused on the Arp8 subunit of INO80. Follow-up studies with multiple independent assays, both in vitro and in vivo, delineated a role for DDK-dependent phosphorylation of INO80 in nucleosome remodelling and generating the proper nucleosome organisation at replication origins. This had profound consequences for the initiation and duration of replication and for the survival of cells upon replication stress. Nucleosome architecture at replication origins is established during the G1 phase, prior to replication initiation[4,76]. Our data are consistent with the otherwise unexplained finding that DDK interacts with ORC at replication origins already in the G1 phase[42,44]. DDK is recruited to early-firing origins during the G1 phase by the forkhead transcription factors Fkh1 and Fkh2[43]. This interaction was crucial for origin function, possibly by competitive recruitment of replication factors from a limited pool. Our data now introduce a further layer of complexity, as we demonstrated that DDK not only targets classical replication factors, but also a chromatin remodeler with nucleosome spacing activity and that this influences replication efficiency (Fig. 5f). Our in vitro replication results

(Fig. 4c, d, e) suggest that replication initiation, rather than elongation, is affected by impaired DDK-dependent phosphorylation of Arp8. Further work is needed to reveal which subpopulation of origins responds to this mechanism in our purified, reconstituted system. It will be interesting to determine the extent to which nucleosome positioning contributes to the firing of an origin early, and the extent to which other chromatin features, such as histone modifications or 3D organisation, play a role. While this remains to be proven, we cautiously speculate that the replication timing programme is finetuned not only at the level of competition between replication origins for the early or late recruitment of replication factors, but also at the level of the early or late establishment of nucleosome organisation at origins conducive to the initiation of replication. Pioneering studies already identified a stimulatory role for remodelers with nucleosome spacing activity, like INO80 or ISW2, in replication in vivo[77,78]. However, the underlying mechanisms remained obscure. More recent biochemical reconstitution studies demonstrated that the proper spacing of nucleosomes was crucial for the process[2,4,79]. The observation that early origins exhibited better-positioned nucleosomes compared to late origins is perfectly consistent with our model[76]. The precise influence of arrays of well-positioned nucleosomes on DNA-templated processes such as transcription or replication remains unknown. It was suggested that closely packed nucleosomes in irregular arrays may impede transcription[80]. The same may be true for replication. There is already evidence that DNA polymerase passage is sensitive to nucleosome positioning, as the lengths of lagging strands appear to correlate with nucleosome positioning[81]. This suggests that regular arrays may play a crucial role in precisely coordinating lagging and leading strand synthesis when replisomes initiate, and nucleosome array regularity may even be more important for replication than for transcription. In accordance with this hypothesis, the *arp8-AA* mutant, in which nucleosome positioning at origins was altered, showed replication defects (Figs. 4 and 5), but minimal changes in transcription (Supplementary Fig. 8).

As discussed above, our data suggest that the INO80 complex and its DDK-dependent phosphorylation play a role in replication initiation rather than elongation. We speculate that the primary function of ATP-dependent chromatin remodelers, such as INO80, is at replication origins, where they generate regularly spaced nucleosomes in conjunction with ORC[4]. These nucleosome patterns are crucial for replication initiation[4]. Once the replisome has escaped the origins, remodelers may become less important, with histone chaperones such as FACT/Nhp6 becoming more important for efficient replication through chromatin[2].

Most intriguingly, the INO80/DDK mechanism appears to play a crucial role in restarting the replication fork following exposure to HU (Fig. 5d). We further speculate that regular, spaced nucleosomal arrays may be formed by an unknown mechanism at stalled replication forks, and that this may be a prerequisite for restarting stalled forks. Previous studies have shown that *arp8* deletion mutants are equally sensitive to MMS and HU[62,82]. In contrast to our work, these studies used *arp8*

deletion mutants, which represent a very different scenario. It is possible that the phosphorylation of Arp8 is important for restarting replication after fork stalling when pools of deoxynucleotide triphosphates (dNTPs) are limited (i.e., following treatment with HU), but not when replication stalls due to DNA alkylation (i.e., following treatment with methyl methanesulfonate (MMS)). Further research is required to clarify this intriguing difference. In contrast to core replication factors, which are solely involved in replication, chromatin factors are involved in all DNA-templated processes, including transcription. Our here reported discovery that cell cycle kinases not only target core replication factors, but also chromatin factors involved in replication[39,83,84], may provide an explanation for the question of how chromatin factors are involved in one DNA-templated process or the other. For our case of the INO80 chromatin remodeler, phosphorylation of Arp8 by DDK may make INO80 "replication competent". The precise mechanism for this remains unclear. Maybe Arp8 phosphorylation is instrumental for the interaction of INO80 with the barrier complex ORC at replication origins[4], while the interaction with other barrier proteins, like Abf1 or Reb1, which cooperate with INO80 in generating phased nucleosome arrays at gene promoters[74], does not depend on Arp8 phosphorylation.

Collectively, our findings demonstrate that DDK not only activates core replication factors but also plays a pivotal role in establishing the nucleosomal landscape at replication origins by targeting the nucleosome spacing remodeler INO80 (Fig. 5f). This provides further insight into the role of DDK at replication origins during G1 phase. We propose that the accurate organisation of nucleosomes is a key factor in replication initiation and may therefore contribute to the replication timing programme. The prevailing view of nucleosomes is no longer limited to their role as obstacles that must be overcome; rather, their proper organisation is now regarded as the gatekeeper of physiological DNA-templated processes, including replication.

## Methods

### Strains and oligos
Yeast strains in this study were generated using standard genetic techniques. The strains were originated from the S288c genetic background. For the generation of INO80 expression strain, a 3X-FLAG was chromosomally inserted at the C-terminus of Ino80 in the indicated strains using pBP83 as a template. All the oligonucleotides were synthetised from Merck. F and R stand for forward and reverse primers, respectively.

Lists of strains and primers used in this study can be found in the Supplementary Information.

### Protein expression and purification
The DDK and ORC complex were expressed and purified as previously described[2,4]. INO80 expression and purification: The INO80 complex was expressed and purified as previously described before with modifications[2,85–89]. The INO80 complex was endogenously expressed, and cells were grown in 16 litres YPD for 24 hours at 30 °C. Cells were collected by centrifugation (6000 x $g$, 10 mins, 4 °C) and pellet was resuspended in an equal volume of 2X lysis buffer with protease inhibitors 0.2 mM PMSF, 1 μM pepstatin A, 1 μg/ml aprotinin, and 2 μM leupeptin. 1X lysis buffer is 25 mM HEPES-KOH pH 7.6, 500 mM KCl, 10% glycerol, 0.05% NP-40, 1 mM EDTA, 1 mM DTT, 4 mM MgCl₂. Cells were frozen in liquid nitrogen in a dropwise manner, and the frozen cells were crushed using a Freezer Mill (SPEX SamplePrep 6875 Freezer/Mill) (6 cycles for 2 min, crushing rate 15). To purify INO80, the frozen cell powder was thawed, resuspended in the 50 ml 1X lysis buffer with protease inhibitors. The insoluble material was cleared by ultracentrifugation ((235,000 x $g$, 1 h, 4 °C). The supernatant was incubated with 1.5 ml pre-washed anti-FLAG M2 affinity gel resin in a batch for 1 h at 4 °C. This was transferred into a disposable column, washed with 4 Column volume (CV) of lysis buffer and 1 CV of wash

buffer (25 mM Tris-HCl pH 7.2, 200 mM KCl, 10% glycerol, 0.05% NP-40, 1 mM EDTA, 1 mM DTT, 4 mM MgCl₂). INO80 was eluted by 1 ml of the same buffer with 0.5 mg/ml 3xFLAG peptide, followed by 2 X 1 ml of buffer with 0.25 mg/ml 3xFLAG peptide. Eluted fractions were analysed by SDS-PAGE, pooled, and further purified with a Mono Q 5/50 GL column using a 15 CV gradient from 100 mM to 600 mM KCl (25 mM Tris-HCl pH 7.2, 5% glycerol, 0.05% NP-40, 1 mM EDTA, 1 mM DTT, 4 mM MgCl₂). The fractions were analysed by SDS-PAGE, pooled, concentrated using a Vivaspin 20. The Concentrated protein was then dialysed for 1.5 h in buffer containing 25 mM Tris-HCl, pH 7.2, 600 mM NaCl, 40% glycerol, 1 mM EDTA and 1 mM DTT.

For cross-linking mass spectroscopy, the purified protein was equilibrated in 25 mM HEPES-NaOH, pH 7.2, 300 mM NaCl, 5% glycerol, 1 mM DTT, 4 mM MgCl₂ using an Amicon Ultra Centrifugal Filter (3 kDa MWCO) for five times.

INO80-AA expression and purification: INO80-AA was purified using yPB5, which contains two point mutations and 3X-FLAG at the C-terminus of the Ino80 subunit. For the protein expression, the INO80 complex was endogenously expressed, and cells were grown in 24 litres YPD for 24 h at 30 °C, collected, processed and purified similar to INO80.

To dephosphorylate INO80 and INO80-AA, purified complexes were treated with Lambda phosphatase at a concentration of 5 μL per ml protein sample for a period of 2 h at 16 °C. This was followed by the addition of alkaline phosphatase at a concentration of 5 μL per 1 ml of protein sample for a period of 6 h at 4 °C. The phosphatases were then removed by rebinding the complexes to FLAG resin and washing them as described above. The complexes were either treated or not treated with DDK and ATP on the beads for one hour at 16 °C. DDK was removed by washing as described above. Subsequently, the complexes were eluted, and the FLAG peptide was removed by MonoQ purification.

### Cell cycle arrest and release
Cells were grown at 30 °C in YPD to OD 600 nm = 0.2–0.5 (OD was measured in a Thermo Scientific GENESYS 20 spectrophotometer). For G1 arrest, cells were treated with alpha-factor – 10 mg/ml stock concentration and 10 μg/ml working concentration- for 120 mins in total. A second dose of alpha-factor was added after 60 mins. For Hydroxyurea arrest, cells were treated with 200 mM Hydroxyurea for 120 minutes. In both cases, cells were then washed with pre-warmed YPD twice and released into fresh pre-warmed YPD to resume the cell cycle. Samples for FACS analyses and mass spectrometry were taken at the indicated time points and processed according to standard protocols.

### Flow cytometry
Cell cycle arrest and release samples were collected as described above. For flow cytometry, samples were processed as described previously[90]. Briefly, yeast cells were harvested by centrifugation, resuspended in 50 mM Tris-HCl, pH 8.0 / 70% ethanol and stored at 4 °C for minimum one hour for permeabilisation and fixation. Cells were then washed once with 50 mM Tris-HCl, pH 8.0 and treated with 200 μg RNase A; diluted in 10 mM Tris-HCl, pH 7.5, 10 mM MgCl₂) for at least 4 h at 37 °C, and subsequently treated with 400 μg Proteinase K (diluted in 10 mM Tris-HCl, pH 7.5) for 30 min at 50 °C. Cells were later resuspended in 50 mM Tris-HCl, pH 8.0, sonicated and diluted 1:20 with 50 mM Tris-HCl, pH 8.0, containing 0.5 μM SYTOX Green and measured with a MACSquant Analyser Flow Cytometer (Milteny Biotec).

### In vitro DDK kinase reactions
15 nM of WT INO80 and mutants or 15 nM MCM complex, were incubated with 5 nM DDK in buffer containing 100 mM potassium-glutamate, 25 mM HEPES-KOH pH 7.6, 10 mM Mg(OAc)₂, 0.02% NP-40, 1 mM DTT, 10 mM ATP and 5 mCi ³²P-g-ATP for 30 mins at 30 °C. Proteins

were then separated on SDS-PAGE, gel were dried, exposed with Super RX Medical X-Ray Film and autoradiograph was developed using a Typhoon scanner phospho imager (GE Healthcare).

## Co-IP experiments

100 nM INO80 nd 150 nM DDK were incubated in buffer containing 100 mM potassium-glutamate, 25 mM HEPES-KOH pH 7.6, 10 mM Mg(OAc)$_2$, 0.02% NP-40, 1 mM DTT, 10 mM ATP for 1 h and then transferred onto pre-washed calmodulin affinity resin (Agilent) in batch at room temperature in the presence of 2 mM CaCl$_2$. While binding to beads, reactions were rotated at room temperature and then washed with the same buffe and 2 mM CaCl$_2$. Proteins were eluted with buffer containing 3 mM EGTA without CaCl$_2$. Supernatants were analysed by immuno-blotting. Antibodies for immunoblots were anti-FLAG M2 peroxidase (Sigma, F 3165; dilution 1: 5000) and anti- CBP (Sigma, 04-932; dilution 1:1000).

## Genetic analysis of recombination

Recombination frequencies were calculated by transforming the yeast strains with the pRS316-L plasmid containing two truncated repeats of the *LEU2* gene sharing 600 bp of homology and placed on a mono-copy *CEN*-based plasmid[91]. Recombinants were obtained by plating appropriate dilutions in SC media lacking leucine and uracil. To calculate total number of cells, they were plated in the same media supplemented with leucine. All plates were grown for 3–7 days at 30 °C. For each transformant, the median value of six independent colonies was obtained. The mean and SEM of at least three independent experiments performed with independent transformants was plotted.

## Spot dilution assay

Cells were grown overnight to near saturation, and OD 600 nm was measured in technical replicates after 1:10 dilutions in water. Cells were diluted to OD 600 nm = 1.0 in 250 μl water, and 5-fold dilutions were generated. Seven microliters of the dilutions were spotted on YPD plates and YPD plates supplemented with the desired drug as necessary. Plates were incubated at 30 ° C for 2 days.

## Transcriptome analysis

For experiments with synchronised cells, strains were grown and treated as above (see "Cell cycle arrest and release"). Otherwise, cells were generally grown overnight to OD 600 nm = 0.8–1.0 in 10 ml YPD complete media. The next day, cells were reinoculated in 100 ml fresh YPAD media at 0.1 OD. At OD 600 nm = 0.4 cells were collected by centrifugation at 4500 x g, 10 mins, 4 °C, the pellets were washed once with cold distilled water, flash frozen in liquid nitrogen and stored at −80 °C.

For the extraction of total RNA, pellets were resuspended in 1 ml QIAzol Lysis Reagent and mixed with 250 μl of zirconia beads. Cells were lysed using the Precellys 24 homogeniser (Bertin) for 3 × 30 sec with 5 min rest in between on ice, followed by centrifugation at 12000 x g, 10 min, 4 °C. To the cleared lysate, 200 μl chloroform was added and mixed by brief vortexing for 15 sec. Samples were incubated at room temperature (RT) for 10 min followed by centrifugation at 12000 x g, 10 min at 4 °C. The aqueous phase was extracted by adding 500 μl chloroform followed by brief vortexing and centrifuged at 12000 x g for 10 min at 4 °C. To the aqueous phase, 500 μl isopropanol was added for 15 min at 4 °C and pelleted by centrifugation at 12000 x g, 10 min, 4 °C to precipitate RNA. The RNA pellets were washed two times with 1 ml of 75% ethanol. The RNA pellet was dissolved in 150 μl RNase-free water for 30 min at 55 °C and quantified photometrically with DeNovix Spectrophotometer. For each sample, 10 μg RNA was treated with 10 units recombinant DNase I for 1 h at 37 °C, purified with Agencourt RNA-Clean XP beads and quantified photometrically with DeNovix Spectrophotometer.

Library preparation and sequencing: For each sample, 1 μg DNaseI-treated RNA was used for rRNA depletion and library preparation. Ribosomal RNA was depleted using the *S. cerevisiae* - specific riboPOOL kit according to the manufacturer's protocol. rRNA-depleted RNA was purified with Agencourt RNAClean XP beads and analysed on the 4150 TapeStation System using a High Sensitivity RNA Screen Tape. Of the rRNA-depleted samples, directional libraries were prepared using the Next Ultra II Directional RNA Library Prep Kit for Illumina in accordance to the recommended protocol. The quality of the libraries was assessed on 4150 TapeStation System using a High Sensitivity D1000 Screen Tape. Libraries were sequenced on an Illumina NextSeq 2000 instrument in paired-end mode.

Data analysis: Sequencing reads were pseudoaligned to the yeast transcriptome (Ensembl build R64, annotation version 108) using kallisto (version 0.48) using default parameters. Data was further processed in R/bioconductor. Differential expression was tested with DESeq2(version 1.36.0) using the experimental batch as a random factor.

## Nuclei isolation

Yeast nuclei were isolated as described previously with modifications[92]. Yeast cells were grown overnight to OD 600 nm = 0.4 – 0.6 in 200 or 500 ml YPD complete media. Cells were harvested by centrifugation at 4500 x g, 10 min, 4 °C and the pellets were washed once with cold distilled water. The weight for washed pellet (wet weight) was determined and resuspended in 2 x wet weight of pre-incubation solution containing 0.7 M β -mercaptoethanol and 2.8 mM EDTA pH 8.0. Cells were shaken for 25 – 30 mins at 30 °C, then washed with 40 ml of cold 1 M sorbitol, and finally the pellet was resuspended 5 ml buffer (5 mM β -mercaptoethanol in 1 M sorbitol) per gm wet weight of the pellet. OD 600 nm of resuspended pellet was determined using a 1:100 dilution in water. To digest the cell wall (spheroblasting), Zymolyase was freshly dissolved in water and was then added to the resuspended pellet. 2 mg Zymolyase was added per gram of wet weight and incubated at 3 °C for 20 – 30 min. The cell wall was digested until the absorbance at 600 nm was decreased to 80 – 90% of the starting OD when the digestion was considered to be complete. Spheroblasts were harvested by centrifugation (2500 x g, 5 min, 4 °C), washed one time with 40 ml cold 1 M sorbitol and resuspended in a Ficoll buffer (18% Ficoll type 400, 20 mM KH$_2$PO$_4$ pH 6.8, 1 mM MgCl$_2$, 0.25 mM EGTA pH 8.0, 0.25 mM EDTA pH 8.0). 7 ml of the Ficoll buffer was added per gram weight of cells. Lastly, nuclei were aliquoted to the desired wet weight (0.5 or 1 g) and centrifuged at 12000 x g, 30 mins, 4 °C. The nuclei pellets were frozen in a dry ice/ethanol bath and were stored at −80 °C until further processing.

## Nuclei processing for mass spectrometry

Mass spectrometry samples for label-free phospho proteome and whole proteome analysis were processed following a published protocol[93]. All solutions were prepared with mass spectrometry-grade reagents. Briefly, thawed nuclei were first washed with 100 mM Tris-HCl, pH 8.5 to remove phosphate buffer residues. Chilled lysis buffer containing 4% Sodium deoxycholate and 100 mM Tris-HCl pH 8.5 was added such that the total volume is about 600 μl. The lysates were heat inactivated at 95 °C for 5 mins and then homogenised by bath-sonication using Biorupter Pico at 4 °C (two times 10 cycles each of 30 sec on and 30 sec off at maximum output power). The cleared lysates were collected in a new eppendorf tube by centrifugation and protein concentration was determined using the Bicinchoninic acid (BCA) protein assay kit. All samples were diluted to a final concentration of 300 μg in 270 μl of lysis buffer. Reduction/alkylation buffer containing 100 mM Tris(2-carboxyethyl) phosphine hydrochloride (TCEP) and 400 mM of 2-chloroacetamide (CAM) was added at a 1:10 ratio of the total volume (30 μl) and incubated for 5 mins, 45 °C with shaking at 1500 rpm. At room temperature, using 1:100 enzyme to

substrate ratio, 3 µg each of Lys-C and trypsin was added to each 300 µg sample and digested overnight at 37 °C with shaking at 1500 rpm.

The next day, the digested peptides were separated such that 2/3rd of the sample was processed for phospho proteome and 1/3rd was processed for whole proteome. Stage tips were prepared as described earlier[94].

For the phospho proteome, phospho peptides were enriched by TiO$_2$ based enrichment and eluted using C8 material on C8 stage tips. Collected phopsho peptides were desalted and eluted using SDB-RPS (Styroldivinylbenzol Reversed Phase Sulfonat) material on SDB-RPS stage tip. The eluate was evaporated under vacuum to dryness at 45 °C. Phospho peptides were then reconstituted in 10 µl of MS loading buffer containing 0.3% Trifluoroactetic acid (TFA) and 2% Acetonitrile (ACN).

For the whole proteome, 1/3rd of the digested peptides were dissolved with 200 µl isopropanol and 1% TFA. Peptides were desalted and eluted using by SDB-RPS material on the SDB-RPS stage tip. The eluate was evaporated under vacuum to dryness at 45 °C and peptides were reconstituted in 15 µl of MS loading buffer.

## Mass spectrometric measurements and data analysis

**LC-MS measurements.** For LC-MS purposes, desalted peptides were injected in an Ultimate 3000 RSLCnano system (Thermo Scientific) and separated in a 25-cm analytical column (75 µm ID, 1.6 µm C18, IonOpticks) with a 50 min gradient from 2 to 35% or 60 min gradient from 2 to 32% ACN in 0.1% formic acid for proteome and phospho-proteome analysis, respectively.

Phospho proteome: The effluent from the HPLC was directly electrosprayed into an Orbitrap Exploris-480 (Thermo Scientific) operated in data-dependent mode to automatically switch between full scan MS and MS/MS acquisition. Survey full scan MS spectra (from m/z 350–1400) were acquired with resolution $R = 60,000$ at 400 m/z (AGC target of $3 \times 10^6$). The 15 most intense peptide ions with charge states between 2 and 5 were sequentially isolated to a target value of $2 \times 10^5$, fragmented at 30% normalised collision energy and acquired with resolution $R = 15,000$. Typical mass spectrometric conditions were: spray voltage, 1.5 kV; no sheath and auxiliary gas flow; heated capillary temperature, 275 °C; ion selection threshold, $5 \times 10^3$ counts. For phosphopeptides, MS2 resolution was increased to $R = 30,000$ and the ion selection threshold to $3 \times 10^4$ counts.

Whole proteome: Eluting peptides were ionised in a nanoESI source and on-line detected on a QExactive HF mass spectrometer (Thermo Fisher Scientific). The mass spectrometer was operated in a TOP10 method in positive ionisation mode, detecting eluting peptide ions in the m/z range from 375 to 1600 and performing MS/MS analysis of up to 10 precursor ions. Peptide ion masses were acquired at a resolution of 60,000 (at 200 m/z). High-energy collision-induced dissociation (HCD) MS/MS spectra were acquired at a resolution of 15,000 (at 200 m/z). All mass spectra were internally calibrated to lock masses from ambient siloxanes. Precursors were selected based on their intensity from all signals with a charge state from 2 + to 5 +, isolated in a 2 m/z window and fragmented using a normalised collision energy of 27%. To prevent repeated fragmentation of the same peptide ion, dynamic exclusion was set to 20 s.

## Database search

MaxQuant search parameters: For phospho proteome identification, the MaxQuant 2.0.3.0 software package was used. Parent ion and fragment mass tolerances were 8 ppm and 0.5 Da, respectively, and allowance for two missed cleavages was made. The Yeast canonical protein database from UniProt (Saccharomyces cerevisiae (strain ATCC 204508 / S288c)), filtered to retain only the reviewed entries were used for the searches. Regular MaxQuant conditions were the

following: site FDR, 0.01; protein FDR, 0.05; minimum peptide length, 6; variable modifications, oxidation (M); phospho (STY); fixed modifications, carbamidomethyl (C); peptides for protein quantitation, razor and unique; minimum peptides, 2; minimum ratio count, 2. Proteins were validated on the basis of at least one unique peptide detected in the proteome of all four replicates or in at least two of the four replicates.

For whole proteome identification, MaxQuant search parameters were identical except for the variable modifications, which were oxidation (M); acetyl (protein N-term); acetyl (K); dimethyl (KR); methyl (KR).

## Data analysis

The phospho-proteomics data was analysed using an R-script developed in-house. Differential and quantitative analysis was performed using phospho-sites with a 75% or higher probability of occurrence (according to MaxQuant output). Phospho-sites and proteins that were present in at least 75% of the replicates were considered for the downstream analysis. Intensity based absolute quantification (iBAQ) values were used to quantify the abundance of phospho-sites and compare it in different conditions. Differential expression analysis at the whole and phospho proteome level was carried out using the DEP package. Briefly, after filtering for all the experimental and analytical contaminants, missing values were imputed by the Bayesian principal component analysis (BPCA) method, followed by limma statistical analysis[95] using a $p$-value cut-off of 0.05. The phospho-site abundances were normalised to the total protein abundance of the respective proteins.

GO term analysis: GO term analysis for the corresponding proteins of the overlapping phosphosites within the DDK active fractions of sc7-4 and Hydroxyurea screens were performed in ImShot[96] using the following parameters: $p$-value cutoff – 0.05, p-value adjustment – Benjamini-Hochberg (BH), Database: org.Sc.sgd.db, Redundancy removed, Minimal and maximal size of genes – 1 and 500, respectively.

## Cross-linking mass spectrometry

Firstly, the equilibrated purified protein was titrated against different DSBU cross-linker concentrations to determine the optimal cross-linking concentration for each protein. The INO80 complex (1 µg in total of INO80 or INO80-AA) was cross-linked for 20 min at 30 °C on a thermomixer at 1200 rpm. The cross-linked product was separated on SDS-PAGE followed by silver staining. The optimal INO80 concentration was determined be 25 µg and 15 µg at 1200 rpm, 20 min, 30 °C. The optimal DSBU cross-linker concentration for INO80 was 58 µM and for the INO80-AA was 100 µM.

The reaction with optimal protein and cross-liner concentration was performed at 1200 rpm, 20 min, 30 °C. The reaction was quenched by adding ammonium bicarbonate to a final concentration of 100 mM and incubated for 10 min at 30 °C, subsequently followed by protein denaturation, alkylation, and tryptic digest. Cross-linked samples were denatured by adding two sample volumes of 8 M urea, reduced with 5 mM Tris (2-carboxyethyl) phosphine (TCEP) and alkylated by the addition of 10 mM iodoacetamide (IAM) for 40 min at RT in the dark. Proteins were digested with 0.5 µg Lys-C at 35 °C for 2 hr, diluted with 50 mM ammonium bicarbonate, and digested with trypsin 0.5 µg overnight. Peptides were acidified with 1% trifluoroacetic acid (TFA) and purified by reversed phase chromatography using C18 material (Empore) on C18 stage tips. Further, cross-linked peptides were enriched on a Superdex Peptide PC 3.2/30 column using water/acetonitrile/TFA (75/25/0.1) as mobile phase at a flow rate of 50 µl/min. Fractions containing cross-linked peptides were analysed by liquid chromatography (Dionex 3000, Thermo Fisher Scientific) coupled to tandem mass spectrometry (LC-MS/MS) using a TimsTOF Pro instrument (Bruker Daltonics).

### LC-MS Analysis of enriched cross-linked peptides

Cross-linked peptides were injected and separated on a PepSep column (25 cm, inner diameter 150 μm, Bruker Daltonics) by an online reversed-phase chromatography with a 50 min gradient from 3 to 43 % of Buffer B containing 100% ACN, 0.1% formic acid) at a flow rate of 300 nl/min. Eluting peptides were directly sprayed through the CSI source into the TimsTOF Pro. Each sample was measured in three independent technical replicates. The mass spectrometric measurement was performed in data-dependent acquisition mode with a top 10 method. The same settings were applied as described[97]. As a template the standard DDA-PASEF MS Method provided by Bruker Daltonics was used. Only precursor ions of + 3 to + 8 charge (in case for DSBU + 2 to + 8 charge) were selected for fragmentation scan. Raw data files were searched against the MaxQuant software package (version 2.0.2.0). The following changes have been applied for the MaxQuant search: Enzyme specificity set to trypsin with a maximum number of missed cleavages 3; DSBU specificity linking (K, S, T, Y); fixed modifications carbamidomethyl (C); variable modifications, oxidation (M). PSM FDR crosslink set to 5%. Inter- and Intra-Crosslinks were filtered by applying an MS1 tolerance window of − 3 to 3 ppm and a score ≥ 60[98]. Cross-links were visualised as network plots using the webserver xiNET[99].

### Salt gradient dialysis (SGD) chromatin

Chromatin in this study was assembled using salt gradient dialysis (SGD) as previously described[4]. Briefly, a yeast origin plasmid library containing ~ 300 ARS (OriDB) sequences was generated from the *S. cerevisiae* genomic library. Selected plasmids contained an origin of replication at least 1000 bp away from the border of the plasmid backbone and yeast genomic insert. Using SGD, 10 μg of origin plasmid library DNA was combined with Drosophila embryo histone in 100 μl SGD buffer containing 10 mM Tris–HCl pH 7.6, 2 M NaCl, 1 mM EDTA pH 8, 20 μg BSA, 0.05% Igepal to a saturated assembly degree. Samples were placed in Slide-ALyzer devices in 300 ml high salt buffer containing 10 mM Tris–HCl pH 7.6, 2 M NaCl, 1 mM EDTA pH 8, 0.05% Igepal, 14.3 mM β-mercaptoethanol). This was dialysed in 3 L of low salt buffer containing 10 mM Tris–HCl pH 7.6, 50 mM NaCl, 1 mM EDTA pH 8, 0.05% NP-40, 1.4 mM β-mercaptoethanol using a peristaltic pump at 7.5 rpm, 30 °C for 16 h. Chromatin was then dialysed for 1 h with 1 L low salt buffer at 30 ° C and stored at 4 ° C.

### Reconstitution and Purification of mono nucleosomes

Windom 601 DNA with 80 base pair extranucleosomal DNA in the 0N80 orientation was used the nucleosome reconstitution as previously described[58]. The DNA fragment of 227 base pair was amplified using Polymerase Chain reaction (PCR), the PCR product was purified using anion exchange chromatography and dialysed in water overnight before being concentrated under vacuum. The purified DNA was then combined with 1.1 fold excess of Drosophila embryo histone in 100 μl SGD buffer containing 25 mM Tris–HCl pH 7.5, 2 M NaCl, 0.25 mM DTT. Samples were placed in Slide-ALyzer devices in 300 ml high salt buffer containing 25 mM Tris–HCl pH 7.5, 2 M NaCl, 0.25 mM DTT. This was dialysed in 3 L of low salt buffer containing 25 mM Tris–HCl pH 7.5, 50 mM NaCl, 0.25 mM DTT using peristaltic pump at 7.5 rpm, 30 °C for 16 h. Mono nucleosomes was then dialysed for 1 h with 1 L low salt buffer at 30 °C. Mono nucleosomes were subsequently purified through anion exchange chromatography using a ResourceQ 1 ml column. Fractions containing nucleosomes were pooled, dialysed to 25 mM Tris–HCl pH 7.5, 50 mM NaCl, 0.25 mM DTT, concentrated and stored at 4 °C.

### Nucleosome mobilisation assay

To monitor the sliding activity of INO80 and INO80-AA, 0N80 nucleosomes with 5′-fluorescein - labelled extranucleosomal DNA were used as described previously[58]. Briefly, 0N80 mononucleosomes (300 nM) were incubated with 30 nM INO80 or INO80-AA in a sliding buffer containing 25 mM HEPES (pH 7.5), 60 mM KCl, 7% glycerol,

0.10 mg/ml bovine serum albumin (BSA), 0.25 mM DTT, and 2 mM MgCl₂ at 25 °C. The sliding reaction was initiated by adding 1 mM ATP and terminated at various time points (15, 30, 45, 60, 120, 300, 600, and 1200 s) by adding 0.2 mg/ml Lambda DNA. Nucleosome species were then separated through native polyacrylamide gel electrophoresis (PAGE) on a 3–12% acrylamide bis-tris gel and visualised using a ChemiDoc imaging system (Bio-Rad). All experiments were performed in triplicate. Gel band quantification was conducted using Image Lab, and the fraction of remodelled bands was plotted as a percentage against reaction time. The data followed a saturation curve and were fitted using an exponential equation in R.

### Electrophoretic mobility shift assay (EMSA)

0N80 mononucleosomes were prepared as described above. 0N80 mononucleosomes were labelled at the 5′end of their extranucleosomal DNA with fluorescein. 300 nM of 0N80 was incubated with increasing concentrations (0 nM, 10 nM, 20 nM, 30 nM, 50 nM and 100 nM) of INO80 and INO80-AA on ice for 30 mins in a buffer containing 25 mM HEPES (pH 7.5), 60 mM KCl, 7% glycerol, 0.25 mM DTT, and 2 mM CaCl₂. Samples were analysed by native PAGE on a 3–12% acrylamide bis-tris gel and visualised using a ChemiDoc imaging system (Bio-Rad). All experiments were performed in triplicates.

### Single molecule DNA Curtains

DNA curtain experiments were carried out as described previously[70] on a prism-type TIRF microscope (Nikon Eclipse Ti2), equipped with three illumination lasers (488, 561 and 640 nm Coherent OBIS), an electron multiplying charged coupled camera (iXon Life, Andor) and a syringe-pump-driven microfluidics system supplying the sample chamber. Custom-made flow cells were assembled from silica-fused slides grafted with chromium barriers produced via E-beam lithography and cover slips with a double-sided tape.

Generation of λ-DNA constructs for DNA curtains: λ-DNA ends were tagged with either biotin or digoxigenin containing oligonucleotides by hybridisation and ligation to cos sites and purified on a HiPrep 16 / 60 Sephacryl S-300 HR column in buffer containing. 10 mM Tris–HCl pH 7.5, 1 mM EDTA and 150 mM NaCl.

Flowcell preparation: Lipid Master Mix was prepared as previously described[70] with modifications. Briefly,100 mg DOPC dissolved in 1 ml chloroform and 1 ml DOPE-PEG and 50 μl DOPE-biotin were mixed and stored at − 20 °C. 100 μl Master Mix was dried using nitrogen gun and applying vacuum for 1–2 hours which is then resolved in 2 ml lipid buffer (10 mM Tris pH 7.5, 200 mM NaCl, 20 mM MgCl2). Lipids were sonicated for 5 cycles with 1 min on and 1 min off with 20% amplitude, filtered through 0.22 μm PVDF filters and stored at 4 °C up to 3 – 4 weeks for subsequent use. Lipids were diluted 1:10 in lipid buffer, and flow cells were incubated three times for 10 min with 200 μl diluted lipids. Following the wash with lipid buffer, flow cells were incubated with 5 μl streptavidin in 1 ml BSA buffer (20 mM HEPES, pH 8.0, 1 mg / ml BSA, 1 mM Mg(OAc)₂). After this, 0.3 pM λ-DNA in BSA buffer was added in four steps with 5 min incubation each time. Single-molecule measurements were performed in INO80 buffer (20 mM HEPES pH 8.0, 1 mg/ml BSA, 4 mM Mg(OAc)₂, 80 mM potassium glutamate, 1 mM ATP, 1 mM DTT). Videos were recorded in NIS Elements (Nikon) and analysed in Igor Pro 8 using custom written code.

DNA curtains binding with INO80-LD555 and INO80-AA-LD555: Labelled INO80 was injected into the flowcell and flushed over the DNA at a constant flow of 0.15 ml/min for at least 100 s to ensure removal of free protein from the flowcell. The flow was then turned off for ~ 2 s to ensure that visible INO80 was bound to the DNA and not adsorbed to the surface, followed by a high flow (1 ml/min) 2 M salt wash to remove bound protein. Afterwards, the procedure was repeated with INO80-AA. The order in which the complexes were flushed in was switched for half of the experiments to remove possible biases. The flow was kept on between steps to keep the amount of DNA at the recorded barrier

stable between injections ansd protein complexes were imaged using a 561 nm laser at 100 mW (0.70 µW/µm2), an illumination time of 50 ms and a frame delay of 1.74 ms.

Data analysis: Binding to DNA was determined in relation to positions of chromium anchors and barriers, and proteins adsorbed to the surface were identified using the short period where the flow was turned off. To calculate the amount of binding events of INO80/INO80-AA to the DNA, individual fluorescent spots were localised as described[100,101] exactly 86.7 s (1700 frames) after opening the injection valve. To reduce noise and increase signal, frames were averaged over 0.259 s (5 frames) before localisation, and five consecutive averaged frames were counted. Because the amount of DNA within one field-of-view can fluctuate strongly between experiments for DNA curtains, the data was normalised by calculating the ratio between INO80 and INO80-AA binding events for each double injection. Ratios were corrected to account for the difference in labelling efficiency between both complexes. Error bars were calculated by resampling the acquired ratios 10000 times and represent the one-sigma limits of the resampled means. The p-value was obtained by applying a two-tailed unpaired t test.

## ATP hydrolysis coupled with NADH oxidation assay

30 nM of the purified INO80 and INO80-AA, were incubated in the assay buffer containing 100 mM potassium-glutamate, 25 mM HEPES-KOH pH 7.6, 10 mM Mg(OAc)$_2$, and 1 mM DTT, supplemented with 3 mM ATP and 3 mM MgCl$_2$, 0.6 mM NADH, 3 mM phosphoenolpyruvate (PEP) and 16 µ/ml lactic dehydrogenase/pyruvate kinase enzymes. 24 µl reactions were prepared in 384 well plates in buffer without ATP and MgCl$_2$. The reactions were tested in sets of 3 technical replicates with two different substrates: PCR amplified Windom 601 DNA with 80 bp extra-nucleosomal DNA at 200 nM (30 ng/µl) and 0N80 reconstituted mono nucleosomes at 150 nM.

ATPase reactions were initiated by adding an ATP / MgCl$_2$ mix to a final concentration of 3 mM each. Plates were incubated at 26 °C in a Biotek PowerWave HT plate reader for 30 mins and absorption at 340 nm was determined every 15 s. For analysis, each time course was fitted to a linear function within a time range where all reactions were linear. From the slope of the reaction and the extinction coefficient of NADH (6220 M$^{-1}$ cm$^{-1}$), the change in NADH concentration was calculated (for 30 µL reactions in Greiner plates, the path length was 0.27273 cm). As oxidation of one NADH equals the hydrolysis of one ATP, the ATP hydrolysis rates were calculated from the slopes.

## In vitro nucleosome positioning assay

The assay was performed with 30 nM ORC and 20 nM INO80 or INO80-AA in a buffer containing 20 mM HEPES pH 7.5, 50 mM NaCl, 3 mM MgCl$_2$, 2.5 mM ATP, 2.5 mM DTT, 0.5 mM EGTA pH 8, 12% glycerol, and started by the addition of SGD chromatin, incubated for 2 h at 30 °C and stopped by addition of 0.2 U apyrase incubated at 30 °C for 20 mins. In order to generate mostly mononucleosomal DNA, the reaction was incubated with 100 U MNase and 1.5 mM CaCl$_2$ for 5 min at 30 °C. The digestion was stopped by the addition of 10 mM EDTA and 0.5% SDS, followed by a proteinase K treatment for 1 hour at 37 °C and ethanol precipitation. Samples were run in 1.5% agarose gels for 1.5 h at 110 V constant in 1X TAE (40 mM Tris, 20 mM acetic acid, 1 mM EDTA), and mononucleosomal DNA was excised and purified using a DNA purification kit.

The sequencing libraries were prepared using 10–50 ng mononucleosomal DNA. The samples were diluted to 10 nM, pooled according to the sequencing reads required (~5 million reads per sample), and quantified via BioAnalyzer. The pool was sequenced with an Illumina NextSeq 1000 in 60 bp paired-end mode (Laboratory for Functional Genome Analysis, Ludwig-Maximilians-Universität Munich). The MNase-seq data was analysed as previously described[4].

## In vitro genome-scale chromatin replication assay

Replication assays using chromatinised DNA were performed as previously described[4], with minor modifications.

SGD chromatin was dialysed in a replication buffer containing 100 mM potassium glutamate, 25 mM HEPES-KOH (pH 7.5), 10 mM MgOAc, 100 µg/ml BSA, 0.02% NP-40, and 1 mM DTT. The SGD chromatin was then incubated with 30 nM ORC and either 20 nM INO80 or the Arp8 mutant for two hours at 30 °C. MCM loading was performed in a buffer containing 25 mM HEPES-KOH (pH 7.5), 100 mM potassium glutamate, 10 mM MgOAc, 0.02% NP-40, 1 mM DTT, 50 nM MCM2-7/Cdt1, and 80 nM Cdc6, which were then incubated for 30 minutes at 30 °C. Subsequently, 50 nM DDK was added, and the mixture was incubated at 30 °C for a further 30 minutes to perform MCM phosphorylation.

For the replication reaction, a master mix containing the following was used: 80 nM Pol α, 10 nM Pol δ, 40 nM Cdc45, 30 nM Dpb11, 20 nM Ctf4, 20 nM Pol ε, 220 nM GINS, 5 nM Mcm10, 50 nM Sld2, 25 nM Sld3/7, 10 nM TopoI, 100 nM RPA, 20 nM S-CDK, 20 nM Csm3/Tof1, 10 nM Mrc1, 20 nM RFC, 20 nM PCNA, 25 nM FACT and 250 nM Nhp6. This was all in a buffer containing 200 mM potassium glutamate, 25 mM HEPES-KOH (pH 7.6), 10 mM magnesium acetate and 1.A solution containing 0.00 µg/ml BSA, 1 mM DTT, 0.01% NP-40, 3 mM ATP, 60 nM [α−32P]dCTP, 20 mM dGTP, dATP and dTTP, as well as 1 mM dCTP and 160 mM CTP, UTP and GTP, was prepared.

The reaction was incubated for the desired amount of time and stopped by adding 50 mM EDTA. Illumina MicroSpin G-50 columns (GE Healthcare) were then used to remove the unincorporated nucleotides. The samples were loaded onto a 0.8% alkaline agarose gel containing 30 mM NaOH and 2 mM EDTA and separated by electrophoresis for 16 h at 24 V.

The gel was fixed in 5% trichloroacetic acid and transferred to Whatman paper. An image was taken with a Typhoon FLA 9500 phosphoimager (GE Healthcare).

## Nuclei isolation and in vivo MNase - seq assay

Nuclei were isolated as described above. The isolated nuclei were digested with MNase in MNase buffer containing 150 mM Tris-HCl, pH 7.5, 500 mM NaCl, 14 mM CaCl2, 2 mM EDTA, 2 mM EGTA and 50 mM β-mercaptoethanol to get a distribution of approximately 80% mononucleosome:20% di-nucleosome. The reaction was stopped by the addition of 1 M Tris-HCl, pH 8.5, 10 mM EDTA and 0.5% SDS. The DNA was purified by proteinase K digestion, phenol-chloroform extraction, ethanol precipitation, RNAse A digestion and isopropanol precipitation. The samples were run in a 1.5% agarose gel, the mononucleosomal DNA was cut out and purified using the GeneJET Gel Extraction kit. Sequencing libraries were prepared and data was analysed as for the in vitro nucleosome positioning assay.

## Reporting summary

Further information on research design is available in the Nature Portfolio Reporting Summary linked to this article.

## Data availability

All data generated or analysed during this study are included in this published article. The raw and processed files from mass spectrometry data (both for Phosphoproteomics and Cross-link MS) have been deposited in the ProteomeXchange Consortium via the PRIDE partner repository[102] with the dataset identifier PXD053426 [http://proteomecentral.proteomexchange.org/cgi/GetDataset?ID=PXD053426]. The raw and processed files from high-throughput sequencing data have been deposited in NCBI's Gene Expression Omnibus (GEO) with the GEO series accession numbers GSE281817 and GSE282419 [https://www.ncbi.nlm.nih.gov/geo/query/acc.cgi?acc=GSE282419]. Source data are provided in this paper.

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

## Acknowledgements

We would like to thank Mariia Likhodeeva and Karl-Peter Hopfner for discussing the results. The authors thank Ignasi Forne of the Protein analysis unit (ZfP) at BioMedical Centre (BMC), LMU, for his help with the LC–MS/MS measurements. Thanks are due to Anuroop Venkatasubramani (AG Imhof) for assistance with data analysis scripts. We also thank John Diffley for the *cdc7-4* stain and Helmut Blum and Stefan Krebs (LAFUGA) for high-throughput sequencing. This work was funded by the Deutsche Forschungsgemeinschaft (DFG) – the German Research Foundation – project ID 213249687 – SFB1064 to P.K., C.F.K. and A.I., and SFB1123 project ID238187445 to A.I. Work in the B.P. laboratory was supported by TU Dortmund University and DFG grants (466479039 and 445098914) awarded to B.P. The work in the A.A Laboratory was supported by the research funding to B.G.G. by Programa Operativo FEDER 2014-2020 and Junta de Andalucía (project US-1380058). Work in the F.M.P. laboratory was supported by DFG grants MU3613/1-2 and 497659230, and in the J.S. laboratory by DFG (374605285) and ERC (758124) grants.

## Author contributions

P.B. optimised the approach for proteome / phosphoproteome and prepared samples for mass spectrometry. P.B. generated the mutant strains, added purification tags to the strains, tested mutant strain/protein using in vivo and in vitro assays, grew cells, purified proteins, optimised purification workflow in accordance with requirements of the downstream assays and performed in vitro kinase and NADH coupled ATP hydrolysis assay, including the analysis and plotting of graphs. P.B. wrote the materials and methods section of the paper and prepared the figures. S.L. analysed and plotted proteome and phosphoproteome datasets. C.K. performed, analysed and plotted the data for cross-linking mass spectrometry experiments. J.F. performed, analysed and plotted in vivo MNase data, performed in vitro replication experiments and in vitro kinase assays. P.B. and L.S. together performed and analysed the assembly and purification of mono-nucleosomes, nucleosome sliding and binding assays. L.G., J.D.B.T., and B.P. performed flow cytometry analysis. E.C. and J.F. performed, analysed and plotted in vitro MNase sequencing. M.A.OB., A.A., and B.G.G. performed and analysed the Recombination assay. M.M. performed RNA sequencing. P.V. and F.M.P. helped in setting up NADH coupled ATP hydrolysis assay. T.S. analysed and plotted RNA seq data. G.L. and J.S. performed single-molecule DNA curtains experiments. C.F.K. performed phosphorylation / dephosphorylation of purified protein and wrote the paper. A.I., B.P. and C.F.K. secured funding, and P.K., B.P., A.I. and C.F.K. contributed intellectually to the paper. All authors were involved in editing.

## Funding

## Competing interests

The authors declare no competing interests
