## [Transparent Peer Review file · Nature Communications]

Dbf4-Dependent Kinase Finetunes INO80 Function at Chromosome Replication Origins

Corresponding Author: Professor Christoph Kurat

Version 0:

Reviewer comments:

Reviewer #1

(Remarks to the Author)

ATP-dependent chromatin remodelers play essential roles in establishing and maintaining nucleosomal positioning and spacing, thereby regulating fundamental DNA metabolic processes. Among these, the INO80 complex has been specifically demonstrated to modulate nucleosome positioning adjacent to replication origins in yeast. In this study, Bansal et al. identified the INO80 complex as one of the targets of DDK through phosphoproteome screening. The authors further explored the potential roles of Arp8 phosphorylation by DDK in maintaining INO80 integrity, regulating nucleosome spacing, initiating replication, and responding to replication stress. However, the study lacks concrete evidence to support the major conclusions presented in the manuscript. At its current stage, the work would be better suited for publication in a more specialized journal to document these findings.

Major concerns:

1. The authors identified 424 potential DDK-dependent targets. How many of these have been validated as direct DDK substrates, both in vivo and in vitro? The authors should consider using the *mcm5-bob1* and *mcm5-bob1cdc7-4ts* mutants to further validate their phosphoproteome analysis results, particularly by collecting S-phase cells for relevant analyses.
2. The results presented in this manuscript do not convincingly support the authors' conclusion that "DDK fine-tunes INO80 function at replication origins." Direct evidence demonstrating that DDK targets INO80 at origin DNA in vivo is necessary to substantiate this claim.
3. The weakest aspect of this study is the assertion regarding the role of Arp8 phosphorylation in stabilizing the integrity of the INO80 complex. The crosslinking mass spectrometry (CXMS) results do not provide sufficient support for this claim. Furthermore, it remains unclear how the observations from CXMS analysis correlate with the various defects associated with INO80-AA.
4. Throughout the paper, the authors primarily investigated the effects of the INO80-AA mutant, comparing it with the INO80 WT complex. This approach is insufficient to conclude that DDK-mediated phosphorylation of INO80 leads to defects in nucleosome positioning and replication initiation. Alternatively, the observed results could reflect how Arp8-AA mutations disrupt or weaken INO80's binding to specific substrates, such as nucleosomes. The authors should assess the affinity of INO80 for nucleosomes before and after DDK phosphorylation. Additionally, generating Arp8-EE/DD phosphomimetic mutants would allow for an examination of their effects on nucleosome binding, positioning, and replication initiation.
5. The impact of DDK phosphorylation on INO80's role in replication initiation remains unclear. The authors should investigate the mechanisms underlying this function of INO80 using the in vitro replication system developed by Kurat et al. in 2017 (Mol. Cell).
6. In Fig. 4d, the differences between +/- DDK conditions for both INO80-WT and INO80-AA are statistically marginal and unlikely to be biologically significant. These small differences could represent experimental variation rather than genuine biological effects.
7. In Fig. 5d, the authors base their model on previous reports of ORC-DDK interaction to connect INO80 activity with replication initiation and timing. However, they do not appear to have experimentally verified these interactions using their

purified components. They should confirm these interactions in their system to support their model.

8. Detailed information regarding DDK concentration and how kinase assays were performed should be provided.

Reviewer #2

(Remarks to the Author)

Bansal et al demonstrates that phosphorylation of INO80 by DDK is critical for maintaining the complex's integrity and functionality both in vitro and in vivo. Using a mass spectrometry-based phosphoproteomic screen, the authors identified around 400 nuclear DDK-dependent phosphorylation sites, including key residues in Arp8 of the INO80 chromatin remodelling complex. Phosphorylation of Arp8 stabilised INO80, enhanced its nucleosome spacing activity, and supported efficient DNA replication and resilience to replication stress, while loss of Arp8 phosphorylation disrupted INO80 function, leading to replication defects and impaired stress response. Neutralisation of the two serine residues on Arp8 (thereby preventing DDK-dependent phosphorylation) demonstrated that accurate nucleosome positioning requires cell cycle regulation of INO80. These findings expand the role of DDK beyond helicase activation, establishing it as a regulator of nucleosome organisation at replication origins and providing evidence that cell cycle kinases directly modulate a nucleosome-spacing remodeller to promote genome stability.

This is a well written manuscript with clear figures and well controlled experiments. The author's discovery that a cell cycle kinase directly modulates a nucleosome remodeller to regulate nucleosome organisation at replication origins is the first of its kind. This exciting observation could be only the first in a new area of research focused on understanding the cell cycle dependence of chromatin remodellers. A further strength of the work is the use of a diverse set of experimental approaches.

I only have a few very minor suggestions on the figures and the text listed below that I believe would further strengthen the work.

Specific Comments:

1. It is interesting that the authors highlight that there are around 400 DDK dependent phosphorylation targets, with many of them being nuclear. In reading the manuscript, I was a little confused why so much emphasis was placed on the INO80 investigation and the rest were not further explored not. The authors could expand on whether some of these 400 targets could be indirect phosphorylation events caused by secondary kinases rather than DDK.

2. Line 195-204: The Arp8 subunit of INO80 is a bona fide DDK target. Fig2D: The inclusion of DDK autophosphorylation as a control shows that the kinase is catalytically active, but may not fully substitute for a positive control substrate. Including a known DDK substrate (e.g., Mcm2, Cdc45, Sld3 or a synthetic peptide) would strengthen the interpretation by demonstrating that DDK can phosphorylate a validated target under the same assay conditions to the same (or similar extent). The authors could add an extended figure or reference another study where the same assay conditions were used for DDK phosphorylation of a known substrate. The authors should also consider identifying the other bands that are not a result of autophosphorylation by DDK in the main figure as they have done in the supplemental figure.

3. Line 247 - 257: Arp8 phosphorylation stimulates INO80 ATPase and nucleosome sliding activity. ATP assay: If the INO80 complex was only partially assembled/misassembled is it possible it might still show an ATP hydrolysis signal without being functional in nucleosome repositioning? It would be helpful if the authors added a comment about the relationship between ATPase activity and functionality.

small corrections for the text:

- Line 50: 'we provide the first evidence for the regulation...'
- Line 70: replace 'into' with 'forming' for clarity
- Line 120: replace 'was' with 'were'
- Line 268: ATPase is spelled incorrectly ('ATPasae')
- Line 284: change the phrasing for better clarity: 'this excludes the possibility that the decreased ATP hydrolysis of the...'
- Line 356: HU is undefined
- Line 1001: 'The next day,' instead of 'Next day the digested'
- Line 1147: capitalise 'Hepes' for consistency
- Line 1191: replace 'KGLu' with 'potassium glutamate' for consistency

Reviewer #3

(Remarks to the Author)

General comment:

Bansal et al. seek to identify, in an unbiased and extensive manner, the nuclear DDK targets in yeast using a phosphoproteome approach. In the second part of the manuscript, they focus on the chromatin remodeller INO80, which is one of the main hits from the phosphoproteome. Using a combination of in vitro and in vivo biochemistry, they investigate the function of the 2 confirmed DDK-dependent phosphosites identified on the INO80 subunit Arp8 (referenced as Arp8-AA). They found out that abrogating the 2 phosphosites does not alter INO80 complex composition but rather has an effect on its

conformation. The authors show that although Arp8-AA still binds DNA and nucleosomes, it exhibits a strong loss of ATPase activity. This is associated to reduction in INO80's nucleosome sliding activity both in vitro and chromatin organization in vivo (even if the effect is more subtle probably due to functional redundancy with other remodellers). In the last part of the manuscript, the authors found out that arp8-AA impacts cell cycle progression, is sensitive to replication stress, and is prone to genome instability.

This study addresses an important and understudied question about the function of DDK during DNA replication. I liked the fact that the authors used two independent experimental setups to identify DDK specific targets (i.e. *cdc7-4* TS mutant and HU treatment) which brings robustness to the results presented. The authors fully acknowledge the remaining limitations (e.g. direct or indirect targets of the DDK complex) that are technically difficult to overcome. The biochemistry has been carefully performed, with state-of-the-art methods and used to thoroughly dissect the consequences associated to the loss of the DDK-dependent phosphorylation on the INO80 complex. In my opinion, the functional analysis of arp8-AA in vivo is the less substantiated part of the manuscript and requires a few additional experiments to fully support the authors conclusions (see major comment).

Major comment:

1. Does the arp8-AA mutant display a replication phenotype? The authors address this question in Fig 5A by cytometry and in Supp Figure 6 by RNA-seq. These two datasets are used by the authors to support the title of the last section "DDK-dependent phosphorylation of INO80 is required for replication and during replication stress in vivo". In my opinion, with the results presented, it is not possible to conclude on a potential replication defect in arp8-AA. Indeed, the cytometry experiment shown in Fig 5A evaluates the ability of arp8-AA to cycle from G1 to S after synchronization with alpha-factor. In the profile presented in Fig 5A, it is clear that arp8-AA shows a significant block in G1 (1C peak is maintained very high until at least 50 min post-release and only start to decrease after 60 min release from alpha-factor). The cytometry profile does not show an enrichment of cells into S phase which precludes any conclusions regarding a potential DNA replication defect. Since INO80 is a chromatin remodeller that may impact genes expression, it is possible that a subset of genes required for the G1/S transition is not regulated properly. The authors try to discard this possibility by performing RNA-seq analyses (shown in supp figure 6), which is good. Their results show that arp8-AA does not show any major transcriptome alterations, and at least not genes of interests (cell cycle and DNA replication genes, as shown in Supp Fig 6). Although this seems to be in agreement with a minimal effect of arp8-AA on gene expression, the RNA-seq has been performed in exponentially growing cells, a setup for which the authors do not observe any cell cycle profile differences when comparing WT and arp8-AA (Fig 5A, async condition). If some genes, important for the G1/S transition are misregulated in arp8-AA, it might only be visible at specific stages of the cell cycle. Thus, the expression analyses should be done in synchronous populations. For instance, by comparing G1 vs release into S phase + 20 min or 30 min. Overall, if the authors want to keep the conclusion as it is, they would need to perform extra-experiments:

- a. directly measure replication fork elongation by the method of their choice in a context where they can discard the G1/S transition effect/delay.
- b. Exclude a transcriptional effect by doing RNA-seq on synchronized cells or if not possible, at least by RT-qPCR or Western Blot to compare cyclins expression kinetics at the G1/S transition in the mutant vs WT.
- c. Measure how cells respond to acute exposure to HU. For instance, is the mutant able to properly restart cell cycle / DNA replication after a short exposure to HU (2-3h), this could be achieved by cytometry or other methods.

Minor comments:

1. Introduction, line 155: I would replace "DNA damage checkpoint" by "DNA replication checkpoint".
2. Results, The authors used two methods to inactivate DDK : *cdc7-4* TS and HU ; why not use the same time for DDK inactivation ? 2h for TS vs 3h for HU.
3. Results, Fig 4B vs Fig4D and page 13, lines 326-329. The authors state that the lambda phosphatase treatment on wild-type INO80 (Fig 4D, INO80 + DDK) recapitulates the nucleosome positioning defect found in INO80-AA (Fig 4B., INO80-AA). I agree that the linker lengths are longer in both condition but the -1/+1 peak intensity is not different +/- DDK phosphorylation (Fig 4D) contrary to INO80-AA vs WT (Fig4B). In addition, the nucleosome free region centered on the ACS is less pronounced without DDK phosphorylation (Fig 4D) whereas I do not see any effect when comparing INO80-AA and INO80 (Fig 4B). Since variations in nucleosomes profiles are subtle, and seems to be different according to the experimental setup, I suggest to rewrite more precisely the description of the results / conclusion to fit better to the corresponding figure.
4. Results, Fig 5C. The authors shows that arp8-AA is sensitive to hydroxyurea but not to MMS as scored by drop assay. It has been shown previously that arp8delta strains are equally sensitive to HU and MMS (see for instance, Van Attikum et al., 2004 or Brahma et al., 2018). I'm not questioning the results, and I totally acknowledge that arp8-AA is not a KO of arp8 but the authors should discuss that point in their manuscript. For instance, are Arp8 S65 and S233 not phosphorylated in response to MMS ? Is it expected that the DDK activities/functions differ between HU and MMS ? In the same line, it would be informative to see how the arp8-AA mutant behave on drop assay with lower doses of HU and higher doses of MMS.
5. Discussion page 16, lines 403-406 and again on page 18, lines 436-438. Do the authors implicate that DDK phosphorylation on INO80 contributes to defining the spatio-temporal program of origin firing ? There is no data in the manuscript that substantiate this claim. But measuring if arp8-AA alter the chromatin binding of INO80 at origins or if origin firing is changed would be something that will help the authors to reinforce their conclusions about a direct function of these two DDK-dependent phosphorylation sites on Arp8 during DNA replication (see major point).
6. Discussion page 17, lines 419-421 : missing word ? I do not fully agree on that since it is not clear if arp8-AA exhibit replication defects, G1/S transition defects or both ?
7. Figure 5a legend: cell cycle analysis by PI staining (or other dye) is not FACS but flow-cytometry, please correct.
8. Resolution of Extended data figure 2 is poor on my PDF version, please adjust for the final version.

Version 1:

Reviewer comments:

Reviewer #1

(Remarks to the Author)

The revised manuscript fails to address the core issue regarding the direct role of DDK phosphorylation on INO80 activity. The authors' conclusions rely entirely on comparing the INO80-AA mutant to the wild-type complex. This approach can only demonstrate that Ser65 and Ser233 of Arp8 are important for INO80's structural integrity and function; it cannot establish a direct regulatory role for DDK phosphorylation of Arp8.

To substantiate their claim, it is essential to perform their key assays with INO80 in the presence and absence of active DDK related to Figures 2 and 3. Only a direct comparison between INO80 and DDK-phosphorylated INO80 can determine whether DDK phosphorylation directly modulates INO80's ATPase and nucleosome remodeling activities.

Reviewer #2

(Remarks to the Author)

The authors have address all my comments completely. This paper is now ready for publication.

Reviewer #3

(Remarks to the Author)

In response to my comments, the authors provided new experimental results that reinforce the message of the manuscript. I am recommending the manuscript for publication without hesitation.

The only information missing from the manuscript is a list of the genes used in the RNA-seq analyses for 'cell cycle associated genes' and 'DNA replication associated genes'. It is important that readers have access to this information.

We would like to thank all three reviewers for taking the time to review our manuscript. We are delighted that the evaluations of our paper were overall very positive. We have addressed all of the reviewers' points by conducting new experiments and providing more detailed clarifications, which have certainly improved the paper. In particular, the reviewers' comments prompted us to perform genome-scale replication reactions *in vitro*, as well as more rigorous *in vivo* experiments. These experiments helped us to reinforce our conclusion that origin-flanking arrays of regularly spaced nucleosomes, generated by the INO80-DDK mechanism, are functional and crucial for efficient replication *in vitro* and *in vivo*.

We added the following new experiments:

1. Co-IPs showing that purified INO80 and DDK physically interact (new ED Fig. 3a).
2. *In vitro* kinase assay with the MCM complex as a positive control (new ED Fig. 3b).
3. Nucleosome binding assays (EMSAs) with INO80 plus/minus DDK phosphorylation *in vitro* (new ED Fig. 6c and d).
4. Genome-scale *in vitro* replication reactions on chromatin assembled with INO80 wild type versus INO80-AA (new Fig. 4c, d and e).
5. *In vivo* FACS analyses of WT versus *arp8-AA* mutant following acute HU block (new Fig. 5d).
6. Transcriptome analyses of WT versus *arp8-AA* mutant during G1 phase and early S phase (new Fig. 5e).

In the following, please find a detailed response to all reviewer comments. To help to navigate through the revisions, we highlighted all direct citations from our revised text as well as new or altered Figures in green.

Reviewer #1 (Remarks to the Author):

ATP-dependent chromatin remodelers play essential roles in establishing and maintaining nucleosomal positioning and spacing, thereby regulating fundamental DNA metabolic processes. Among these, the INO80 complex has been specifically demonstrated to modulate nucleosome positioning adjacent to replication origins in yeast. In this study, Bansal et al. identified the INO80 complex as one of the targets of DDK through phosphoproteome screening. The authors further explored the potential roles of Arp8 phosphorylation by DDK in maintaining INO80 integrity, regulating nucleosome spacing, initiating replication, and responding to replication stress. However, the study lacks concrete evidence to support the major conclusions presented in the manuscript. At its current stage, the work would be better suited for publication in a more specialized journal to document these findings.

Major concerns:

1. The authors identified 424 potential DDK-dependent targets. How many of these have been validated as direct DDK substrates, both *in vivo* and *in vitro*? The authors should consider using the *mcm5-bob1* and *mcm5-bob1cdc7-4ts* mutants to further validate their phosphoproteome analysis results, particularly by collecting S-phase cells for relevant analyses.

Unlike CDK targets, few DDK targets have been characterised to date, which is why we screened for them. One of the few substrates that has been studied extensively is MCM, which we also identified in our screening process, thereby validating its ability to detect DDK targets (Fig. ED 1). As with all large-scale screening approaches, both of our screens have limitations, which we discuss at the beginning of the 'Discussion' section of the revised manuscript. The *mcm5-bob1-1* mutant suggested by the reviewer has been used for phosphoproteomic analysis of DDK substrates outside S phase (Galanti et al., 2024), but, as this mutant also affects S phase, it is impossible to distinguish direct from indirect phosphorylation events. The same is true for the *cdc7-4* mutant. We therefore used an *in vitro* approach with purified DDK kinase, which is free of such limitations, to demonstrate the ability of DDK to directly phosphorylate Arp8 and known subunits of the MCM complex (new Figure ED 3b and point 8). While we appreciate the reviewer's suggestion to use these mutants in future studies, we are not convinced that the results of another screen would provide additional support for our current conclusions.

2. The results presented in this manuscript do not convincingly support the authors' conclusion that "DDK fine-tunes INO80 function at replication origins." Direct evidence demonstrating that DDK targets INO80 at origin DNA *in vivo* is necessary to substantiate this claim.

We thank the reviewer for pointing this out. The localisation and function of DDK at origins is well documented (e.g. activation of the MCM complex). We and others have demonstrated that INO80 is functionally active at origins. However, we did not include citations about the localisation of INO80 to origins, although this has been demonstrated by many labs. We apologise for this oversight and we now write (page 5, line 123: "*INO80 localises to replication origins in vivo*⁴⁵⁻⁴⁷, and we previously identified a critical INO80 function in establishing nucleosome arrays at replication origins⁴. This led us to wonder whether DDK-dependent Arp8 phosphorylation might regulate this INO80 function."

We believe that the reviewer may also refer to the absence of evidence showing that the INO80-DDK mechanism primarily affects early replication origins and assists in establishing the replication timing programme. We agree that this is an important point and needs clarification. We now have new data showing that, in a purified, reconstituted system using a library of ~300 yeast origins (see Chacin et al., 2023), this mechanism plays a key role in initiating replication from a subset of origins (new Fig 4c, d and e, which were requested by the reviewer in point 5). This is consistent with our *in vivo* data showing a delay in entry into S phase in *arp8-AA* mutants compared to wild type (WT). The mutants also accumulate a significant fraction of G1 phase cells. We can rule out that this is caused by deregulated transcription of replication and cell cycle genes during the G1 and early S phases (new Fig. 5e). Together with our *in vitro* data, this suggests that a subpopulation of origins has difficulty initiating replication when Arp8 is not phosphorylated by DDK. We are currently developing methods to study this question in our *in vitro* system, such as identifying which fraction of origins is affected and compare that to *in vivo* data. This is very exciting but clearly out of the scope of the current manuscript. We clarified this throughout the manuscript and also modified the model Figure accordingly (new Fig. 5f). We now write (page 18, line 455): "*Our in vitro replication results (Fig. 4c, d and e) suggest that replication initiation, rather than elongation, is affected by impaired DDK-dependent phosphorylation of Arp8. Further work is needed to reveal which subpopulation of origins respond to this mechanism in our purified, reconstituted system. It will be interesting to determine the extent to which nucleosome positioning contributes to the firing of an origin early, and the extent to which other chromatin features, such as histone modifications or 3D organisation, play a role.*

While this remains to be proven, we cautiously speculate that the replication timing programme is fine-tuned not only at the level of competition between replication origins for the early or late recruitment of replication factors, but also at the level of the early or late establishment of nucleosome organisation at origins conducive to the initiation of replication.”; and (page 21, line 512): “We propose that the accurate organisation of nucleosomes is a key factor in replication initiation and may therefore contribute to the replication timing programme.”

We hope the reviewer will concur that this has enhanced the clarity of the manuscript and will maintain readers' enthusiasm for future developments in this area.

3. The weakest aspect of this study is the assertion regarding the role of Arp8 phosphorylation in stabilizing the integrity of the INO80 complex. The crosslinking mass spectrometry (CXMS) results do not provide sufficient support for this claim. Furthermore, it remains unclear how the observations from CXMS analysis correlate with the various defects associated with INO80-AA.

We agree that, if considered in isolation, the experiments conducted by CXMS on INO80 and INO80-AA do not exhaustively answer how Arp8 phosphorylation affects INO80 structure. However, our conclusion that Arp8 phosphorylation affects INO80 activity is mainly based on our many other results (both *in vivo* and *in vitro*). The CXMS results further support this, but need not provide comprehensive elucidation of the mechanism. Subsequent studies that employ high-resolution methodologies (e. g. cryo-EM) in conjunction with INO80 versus INO80-AA bound to a nucleosome are needed for this but fall outside the scope of the present manuscript. In any case, we were struck by the fact that eliminating two amino acids from an intrinsically disordered region had such a profound effect on the overall architecture of the INO80 complex and think it is worthwhile to already report this in the present manuscript.

4. Throughout the paper, the authors primarily investigated the effects of the INO80-AA mutant, comparing it with the INO80 WT complex. This approach is insufficient to conclude that DDK-mediated phosphorylation of INO80 leads to defects in nucleosome positioning and replication initiation. Alternatively, the observed results could reflect how Arp8-AA mutations disrupt or weaken INO80's binding to specific substrates, such as nucleosomes. The authors should assess the affinity of INO80 for nucleosomes before and after DDK phosphorylation. Additionally, generating Arp8-EE/DD phosphomimetic mutants would allow for an examination of their effects on nucleosome binding, positioning, and replication initiation.

We acknowledge that the serine-to-alanine mutation does not necessarily demonstrate that the phosphorylation of INO80 by DDK influences the nucleosome positioning function. We therefore phosphorylated INO80 *in vitro* by DDK and tested the effect on nucleosome positioning around origins in our origin library reconstitution assay (ED Fig. 6a and b). This showed indeed that phosphorylation by DDK is important for the nucleosome positioning function of INO80. In this context, we concur with the reviewer that we should also control for effects on nucleosome binding. We now added this control (EMSA assays) and confirmed that the phosphorylation status of INO80 does not make a difference in nucleosome binding (new ED Fig. 6c and d). We now write (page 14, line 343): “*Defects in nucleosome positioning when using the dephosphorylated INO80 complex may be due to altered nucleosome binding compared with the phosphorylated complex. We therefore tested the ability of the differentially phosphorylated INO80 complex to bind mononucleosomes using EMSAs. The results revealed*

similar nucleosome binding for both the unphosphorylated and phosphorylated INO80 complexes (Extended data Fig. 6 c and d). These findings support our hypothesis that DDK phosphorylation plays a direct role in INO80-mediated nucleosome organisation rather than regulating its affinity to nucleosomes.”

These biochemical experiments clearly demonstrate the importance of phosphorylation for this mechanism and obviate the need to further employ phospho-mimetic mutations. Our direct phosphorylation approach may even be considered as more relevant as it does not involve amino acid substitutions, which may lead to confounding side effects.

5. The impact of DDK phosphorylation on INO80's role in replication initiation remains unclear. The authors should investigate the mechanisms underlying this function of INO80 using the *in vitro* replication system developed by Kurat et al. in 2017 (Mol. Cell).

We thank the reviewer for suggesting these experiments. As discussed in point 2 above, we performed *in vitro* replication assays (new Fig. 4c, d and e). These results serve to reinforce the conclusion that nucleosomal spacing by the INO80-DDK mechanism exerts a direct influence on replication initiation. We now write (page 14, line 351): *“Previously, we demonstrated that precise nucleosome positioning at replication origins is essential for replication initiation⁴. We therefore investigated whether defects in nucleosomal architecture at replication origins, as observed with the INO80-AA mutant complex (Fig. 4b), might cause replication problems. We tested this using our genome-scale, in vitro chromatin replication system⁴ (Fig. 4c). Robust replication was observed when chromatin was assembled using wild-type INO80 complex (Fig. 4d, lanes 1 and 2) but was significantly reduced when we used the INO80-AA mutant complex (Fig. 4d, lanes 3 and 4). Interestingly, the sizes of the leading strands were similar in both experiments, suggesting that replication initiation from a subpopulation of origins was defective (Fig. 4e).*

Taken together, our biochemical assays provide direct evidence that DDK-dependent phosphorylation of the Arp8 subunit of INO80 is involved in its role in nucleosomal spacing. Furthermore, they suggested that this leads directly to defects in replication initiation, rather than elongation.”

6. In Fig. 4d, the differences between +/- DDK conditions for both INO80-WT and INO80-AA are statistically marginal and unlikely to be biologically significant. These small differences could represent experimental variation rather than genuine biological effects.

Yes, the defects in nucleosome positioning are relatively small *in vivo* and we discuss this in the paper (page 15, line 372): *“The arp8-AA mutant showed weaker, but reproducible dampening of array regularity compared to an isogenic ARP8 strain than the respective differences in our in vitro assays (Fig. 4b). This may argue for some redundancy in vivo that we did not reconstitute in vitro”*. *In vivo* nucleosome positioning data is sometimes difficult to interpret in isolation, particularly when the effects are small. Therefore, it was imperative to us to analyse this using a well-defined biochemical system too. We believe that this combination is highly effective, and we are therefore confident in the relevance of the *in vivo* data.

7. In Fig. 5d, the authors base their model on previous reports of ORC-DDK interaction to connect INO80 activity with replication initiation and timing. However, they do not appear to have experimentally verified these interactions using their purified components. They should confirm these interactions in their system to support their model.

As demonstrated in our previous study (Chacin et al., 2023), INO80 has been shown to interact with ORC. Furthermore, the results of our *in vitro* kinase assay have indicated a necessity for interaction between DDK and INO80. However, as requested, we now also performed Co-IP experiments with purified INO80 and DDK and directly verified this interaction (new ED Fig. 3a).

8. Detailed information regarding DDK concentration and how kinase assays were performed should be provided.

As outlined in the Method section, all protein concentrations and buffer conditions are listed. Furthermore, in order to demonstrate the specificity of the assay, *in vitro* kinase assays were conducted with DDK and its well-documented substrate, MCM (as requested by Reviewer 2; new ED Fig. 3b). We now write (page 8, line 200): *“In a pilot experiment, we used the assay to phosphorylate DDK’s known target: the MCM complex. The Mcm4 and Mcm6 subunits, both of which are confirmed major DDK targets^{32,49,54-56}, were clearly phosphorylated by DDK in vitro (Extended data Fig. 3b). The other subunits were not phosphorylated, demonstrating the specificity of our assay.”*

Reviewer #2 (Remarks to the Author):

We are delighted that the reviewer finds: *“This exciting observation could be only the first in a new area of research focused on understanding the cell cycle dependence of chromatin remodellers.”*

Bansal et al demonstrates that phosphorylation of INO80 by DDK is critical for maintaining the complex’s integrity and functionality both *in vitro* and *in vivo*. Using a mass spectrometry-based phosphoproteomic screen, the authors identified around 400 nuclear DDK-dependent phosphorylation sites, including key residues in Arp8 of the INO80 chromatin remodelling complex. Phosphorylation of Arp8 stabilised INO80, enhanced its nucleosome spacing activity, and supported efficient DNA replication and resilience to replication stress, while loss of Arp8 phosphorylation disrupted INO80 function, leading to replication defects and impaired stress response. Neutralisation of the two serine residues on Arp8 (thereby preventing DDK-dependent phosphorylation) demonstrated that accurate nucleosome positioning requires cell cycle regulation of INO80. These findings expand the role of DDK beyond helicase activation, establishing it as a regulator of nucleosome organisation at replication origins and providing evidence that cell cycle kinases directly modulate a nucleosome-spacing remodeller to promote genome stability.

This is a well written manuscript with clear figures and well controlled experiments. The author's discovery that a cell cycle kinase directly modulates a nucleosome remodeller to regulate nucleosome organisation at replication origins is the first of its kind. This exciting observation could be only the first in a new area of research focused on understanding the cell cycle dependence of chromatin remodellers. A further strength of the work is the use of a diverse set of experimental approaches.

I only have a few very minor suggestions on the figures and the text listed below that I believe would further strengthen the work.

Specific Comments:

1. It is interesting that the authors highlight that there are around 400 DDK dependent phosphorylation targets, with many of them being nuclear. In reading the manuscript, I was a little confused why so much emphasis was placed on the INO80 investigation and the rest were not further explored not. The authors could expand on whether some of these 400 targets could be indirect phosphorylation events caused by secondary kinases rather than DDK.

As described in our previous publication (Chacin et al., 2023), INO80 plays a crucial role in shaping the chromatin landscape at replication origins. This is why our focus was on INO80. However, we acknowledge that there are many more potential DDK targets that could be explored in future studies. To emphasise this point, we have added a new network figure (new ED Fig. 2). In the paper, we also acknowledge that some candidates may not have been phosphorylated primarily by DDK. While we agree with all the points raised by the reviewer, at this stage our focus was on INO80's role after DDK phosphorylation. We hope this will generate excitement about future developments in this area!

2. Line 195-204: The Arp8 subunit of INO80 is a bona fide DDK target. Fig2D: The inclusion of DDK autophosphorylation as a control shows that the kinase is catalytically active, but may not fully substitute for a positive control substrate. Including a known DDK substrate (e.g., Mcm2, Cdc45, Sld3 or a synthetic peptide) would strengthen the interpretation by demonstrating that DDK can phosphorylate a validated target under the same assay conditions to the same (or similar extent). The authors could add an extended figure or reference another study where the same assay conditions were used for DDK phosphorylation of a known substrate. The authors should also consider identifying the other bands that are not a result of autophosphorylation by DDK in the main figure as they have done in the supplemental figure.

We acknowledge that we did not include a positive control in our *in vitro* kinase assays. However, we have now included some new data in which we have replicated chromatinised origin libraries generated by INO80 and INO80-AA *in vitro* (new Fig. 4c, d and e; requested by reviewers no. 1 and 3). On the one hand, this really drives home our claim that the INO80-DDK mechanism is important for replication in a purified system, and corroborates our *in vivo* data nicely. On the other hand, this assay demonstrates that the MCM complex is phosphorylated by DDK, which is an essential step in the assay.

However, at the reviewer's request, we also applied our *in vitro* kinase assay to the MCM complex (to the best of our knowledge, Cdc45 and Sld3 are CDK substrates in yeast, but not DDK substrates). We now write (page 8, line 200): *"In a pilot experiment, we used the assay to phosphorylate DDK's known target: the MCM complex. The Mcm4 and Mcm6 subunits, both of which are confirmed major DDK targets^{32,49,54-56}, were clearly phosphorylated by DDK in vitro (Extended data Fig. 3b). The other subunits were not phosphorylated, demonstrating the specificity of our assay."*

3. Line 247 - 257: Arp8 phosphorylation stimulates INO80 ATPase and nucleosome sliding activity. ATP assay: If the INO80 complex was only partially assembled/misassembled is it possible it might still show an ATP hydrolysis signal without being functional in nucleosome repositioning? It would be helpful if the authors added a comment about the relationship between ATPase activity and functionality.

In principle, ATP hydrolysis rates of a chromatin remodeler may become uncoupled from nucleosome remodeling rates (Clapier et al., 2016), i.e., hydrolysis rate remains high or is even increased while sliding rates are decreased. However, in our case both the hydrolysis

and the sliding rates were decreased if Arp8 was not phosphorylated so that the relationship between ATPase activity and functionality was directly correlated.

small corrections for the text:

We would like to express our gratitude to the reviewer for highlighting these errors. All errors were corrected.

- Line 50: 'we provide the first evidence for the regulation...'
- Line 70: replace 'into' with 'forming' for clarity
- Line 120: replace 'was' with 'were'
- Line 268: ATPase is spelled incorrectly ('ATPasae')
- Line 284: change the phrasing for better clarity: 'this excludes the possibility that the decreased ATP hydrolysis of the...'
- Line 356: HU is undefined
- Line 1001: 'The next day,' instead of 'Next day the digested'
- Line 1147: capitalise 'Hepes' for consistency
- Line 1191: replace 'KGlu' with 'potassium glutamate' for consistency

Reviewer #3 (Remarks to the Author):

We are very happy that the reviewer feels that our work "*...addresses an important and understudied question about the function of DDK during DNA replication.*"

General comment:

Bansal et al. seek to identify, in an unbiased and extensive manner, the nuclear DDK targets in yeast using a phosphoproteome approach. In the second part of the manuscript, they focus on the chromatin remodeller INO80, which is one of the main hits from the phosphoproteome. Using a combination of in vitro and in vivo biochemistry, they investigate the function of the 2 confirmed DDK-dependent phosphosites identified on the INO80 subunit Arp8 (referenced as Arp8-AA). They found out that abrogating the 2 phosphosites does not alter INO80 complex composition but rather has an effect on its conformation. The authors show that although Arp8-AA still binds DNA and nucleosomes, it exhibits a strong loss of ATPase activity. This is associated to reduction in INO80's nucleosome sliding activity both in vitro and chromatin organization in vivo (even if the effect is more subtle probably due to functional redundancy with other remodelers). In the last part of the manuscript, the authors found out that arp8-AA impacts cell cycle progression, is sensitive to replication stress, and is prone to genome instability.

This study addresses an important and understudied question about the function of DDK during DNA replication. I liked the fact that the authors used two independent experimental setups to identify DDK specific targets (i.e. cdc7-4 TS mutant and HU treatment) which brings robustness to the results presented. The authors fully acknowledge the remaining limitations (e.g. direct or indirect targets of the DDK complex) that are technically difficult to overcome. The biochemistry has been carefully performed, with state-of-the-art methods and used to thoroughly dissect the consequences associated to the loss of the DDK-dependent phosphorylation on the INO80 complex. In my opinion, the functional analysis of arp8-AA in vivo is the less substantiated part of the manuscript and requires a few additional experiments

to fully support the authors conclusions (see major comment).

Major comment:

1. Does the arp8-AA mutant display a replication phenotype? The authors address this question in Fig 5A by cytometry and in Supp Figure 6 by RNA-seq. These two datasets are used by the authors to support the title of the last section “DDK-dependent phosphorylation of INO80 is required for replication and during replication stress *in vivo*”. In my opinion, with the results presented, it is not possible to conclude on a potential replication defect in arp8-AA. Indeed, the cytometry experiment shown in Fig 5A evaluates the ability of arp8-AA to cycle from G1 to S after synchronization with alpha-factor. In the profile presented in Fig 5A, it is clear that arp8-AA shows a significant block in G1 (1C peak is maintained very high until at least 50 min post-release and only start to decrease after 60 min release from alpha-factor). The cytometry profile does not show an enrichment of cells into S phase which precludes any conclusions regarding a potential DNA replication defect. Since INO80 is a chromatin remodeller that may impact genes expression, it is possible that a subset of genes required for the G1/S transition is not regulated properly. The authors try to discard this possibility by performing RNA-seq analyses (shown in supp figure 6), which is good. Their results show that arp8-AA does not show any major transcriptome alterations, and at least not genes of interests (cell cycle and DNA replication genes, as shown in Supp Fig 6). Although this seems to be in agreement with a minimal effect of arp8-AA on gene expression, the RNA-seq has been performed in exponentially growing cells, a setup for which the authors do not observe any cell cycle profile differences when comparing WT and arp8-AA (Fig 5A, async condition). If some genes, important for the G1/S transition are misregulated in arp8-AA, it might only be visible at specific stages of the cell cycle. Thus, the expression analyses should be done in synchronous populations. For instance, by comparing G1 vs release into S phase + 20 min or 30 min. Overall, if the authors want to keep the conclusion as it is, they would need to perform extra-experiments:

We are very grateful for the thoughtful comments of the reviewer and have conducted all the recommended experiments. We firmly believe that incorporating these new data greatly enhances our paper.

a. directly measure replication fork elongation by the method of their choice in a context where they can discard the G1/S transition effect/delay.

We agree that our *in vivo* replication analyses alone cannot rule out any contribution from the cell cycle machinery. We cannot determine whether any defects are caused by replication and, if so, whether this is due to defective initiation or elongation. In our opinion, the best way to demonstrate this directly is to use our genome-scale *in vitro* chromatin replication assay (Chacin et al., 2023). We feel that this is the most powerful and straightforward approach, as it allows us to clearly eliminate any possible indirect effects on the process. We performed this assay on templates where a library consisting of ~ 300 origins was chromatinised with INO80 and INO80-AA and replication was conducted with purified factors.

We now write: (page 14, lane 351): “Previously, we demonstrated that precise nucleosome positioning at replication origins is essential for replication initiation⁴. We therefore investigated whether defects in nucleosomal architecture at replication origins, as observed with the INO80-AA mutant complex (Fig. 4b), might cause replication problems. We tested this using our genome-scale, *in vitro* chromatin replication system⁴ (Fig. 4c). Robust replication was observed when chromatin was assembled using wild-type INO80 complex (Fig.

4d, lanes 1 and 2) but was significantly reduced when we used the INO80-AA mutant complex (Fig. 4d, lanes 3 and 4). Interestingly, the sizes of the leading strands were similar in both experiments, suggesting that replication initiation from a subpopulation of origins was defective (Fig. 4e).

Taken together, our biochemical assays provide direct evidence that DDK-dependent phosphorylation of the Arp8 subunit of INO80 is involved in its role in nucleosomal spacing. Furthermore, they suggested that this leads directly to defects in replication initiation, rather than elongation.”

These results clearly demonstrate that the replication problem observed in our *in vivo* data is indeed caused by misaligned nucleosomes at replication origins, rather than by deregulated transcription, for example. This is further corroborated by our new RNA-seq data on synchronised cells (see point (b)). It is also encouraging that the *in vitro* data are consistent with some of the flow cytometry profile phenomena.

We can now conclude that (page 15, 378): “We observed an increased block in G1, where non-replicated DNA (1C) remained at high levels for up to 50 minutes after release from α -factor, only decreasing after 60 minutes. As cells did not accumulate in S phase, we could not distinguish between elongation and initiation using this assay. Nevertheless, the data are consistent with our *in vitro* replication results, which showed that initiation, rather than elongation, was affected when chromatin was assembled with the INO80-AA mutant complex (Fig. 4c, d and e).”

b. Exclude a transcriptional effect by doing RNA-seq on synchronized cells or if not possible, at least by RT-qPCR or Western Blot to compare cyclins expression kinetics at the G1/S transition in the mutant vs WT.

These are excellent experiments, and we thank the reviewer for highlighting them. We performed these RNA-seq experiments with synchronised cells, analysing the G1 and early S phases in particular. We now write (page 16, line 403): “We reasoned that, if transcription of genes involved in the G1 to S transition were affected in the *arp8-AA* mutant, this would only be visible at specific stages of the cell cycle. Consequently, we examined the transcriptome using RNA sequencing of synchronised cells released into the G1 phase and the early S phase (Fig. 5e). No significant disparities in the transcription of cell cycle or replication genes were observed at either stage. The only two genes that were deregulated were metabolic genes, which were unlikely to cause a delay in the cell cycle. Thus, these analyses provide further support for our model, which proposes that replication defects in the *arp8-AA* mutant was caused by misaligned nucleosomes at replication origins rather than a failure in transcription regulation.”

c. Measure how cells respond to acute exposure to HU. For instance, is the mutant able to properly restart cell cycle / DNA replication after a short exposure to HU (2-3h), this could be achieved by cytometry or other methods.

Yes, because the *arp8-AA* cells exhibit strong growth defect when cultured on HU, so this is a great suggestion. We have now performed these experiments and can report the following (page 16, line 392): “We thus speculate that Arp8 phosphorylation is crucial for restarting replication after replisomes have stalled due to limited NTP pools (i.e. in the presence of HU). We therefore analysed how *arp8-AA* responded to acute exposure to HU (Fig.

5d). Following a 2-hour exposure to HU, arp8-AA cells were able to restart replication when released from the HU block, albeit with a significant delay compared to the wild type. Whether this restart defect was due to elongation defects (from already initiated forks) or initiation cannot be discriminated by these experiments.”

Minor comments:

1. Introduction, line 155: I would replace “DNA damage checkpoint” by “DNA replication checkpoint”.

Agreed - we have changed that.

2. Results, The authors used two methods to inactivate DDK : cdc7-4 TS and HU ; why not use the same time for DDK inactivation ? 2h for TS vs 3h for HU.

This was the standard procedure when we started conducting these experiments, and we did not change it. We do not think it is necessary to comment on this specifically in the paper, nor do we think this information is necessary in the main figure. We have now omitted this information from the figure, only mentioning it in the methods section.

3. Results, Fig 4B vs Fig4D and page 13, lines 326-329. The authors state that the lambda phosphatase treatment on wild-type INO80 (Fig 4D, INO80 + DDK) recapitulates the nucleosome positioning defect found in INO80-AA (Fig 4B., INO80-AA). I agree that the linker lengths are longer in both condition but the -1/+1 peak intensity is not different +/- DDK phosphorylation (Fig 4D) contrary to INO80-AA vs WT (Fig4B). In addition, the nucleosome free region centered on the ACS is less pronounced without DDK phosphorylation (Fig 4D) whereas I do not see any effect when comparing INO80-AA and INO80 (Fig 4B). Since variations in nucleosomes profiles are subtle, and seems to be different according to the experimental setup, I suggest to rewrite more precisely the description of the results / conclusion to fit better to the corresponding figure.

We agree that this important point needs clarification. We now write (page 13, line 336): *“It should be noted that MNase-seq is very reliable with regard to nucleosome positions (peak positions), but less reliable and reproducible with regard to nucleosome occupancies (peak heights)^{72,73}. Therefore, our conclusions are only based on comparisons of peak positions rather than relative peak heights.”*

4. Results, Fig 5C. The authors shows that arp8-AA is sensitive to hydroxyurea but not to MMS as scored by drop assay. It has been shown previously that arp8delta strains are equally sensitive to HU and MMS (see for instance, Van Attikum et al., 2004 or Brahma et al., 2018). I'm not questioning the results, and I totally acknowledge that arp8-AA is not a KO of arp8 but the authors should discuss that point in their manuscript. For instance, are Arp8 S65 and S233 not phosphorylated in response to MMS ? Is it expected that the DDK activities/functions differ between HU and MMS ? In the same line, it would be informative to see how the arp8-AA mutant behave on drop assay with lower doses of HU and higher doses of MMS.

This is an interesting point, and we thank the reviewer for highlighting it. We now discuss this point in the 'Discussion' section (page 20, line 487): *“Most intriguingly, the*

INO80/DDK mechanism appears to play a crucial role in restarting the replication fork following exposure to HU (Fig. 5d). We further speculate that regular, spaced nucleosomal arrays may be formed by an unknown mechanism at stalled replication forks, and that this may be a prerequisite for restarting stalled forks. Previous studies have shown that arp8 deletion mutants are equally sensitive to MMS and HU^{62,82}. In contrast to our work, these studies used arp8 deletion mutants, which represents a very different scenario. It is possible that the phosphorylation of Arp8 is important for restarting replication after fork stalling when pools of deoxynucleotide triphosphates (dNTPs) are limited (i.e. following treatment with hydroxyurea (HU)), but not when replication stalls due to DNA alkylation (i.e. following treatment with methyl methanesulfonate (MMS)). Further research is required to clarify this intriguing difference.”

5. Discussion page 16, lines 403-406 and again on page 18, lines 436-438. Do the authors implicate that DDK phosphorylation on INO80 contributes to defining the spatio-temporal program of origin firing? There is no data in the manuscript that substantiate this claim. But measuring if arp8-AA alter the chromatin binding of INO80 at origins or if origin firing is changed would be something that will help the authors to reinforce their conclusions about a direct function of these two DDK-dependent phosphorylation sites on Arp8 during DNA replication (see major point).

We agree that this is an important point and needs clarification. As described above, our new *in vitro* replication data suggest that this mechanism plays a key role in initiating replication from a subset of origins (new Fig 4c, d and e). This is consistent with our *in vivo* replication data showing a delay in entry into S phase in *arp8-AA* mutants compared to wild type (WT). The mutants also accumulate a significant fraction of G1 phase cells. We can rule out that this is caused by deregulated transcription of replication and cell cycle genes during the G1 and early S phases (new Fig. 5e). Together with our *in vitro* data, this suggests that a subpopulation of origins has difficulty initiating replication when Arp8 is not phosphorylated by DDK. However, as the reviewer correctly points out, there is no supporting data in the manuscript. We are currently developing methods to study this question in our *in vitro* system, such as identifying which fraction of origins is affected and compare that to *in vivo* data. This is very exciting but clearly out of the scope of the current manuscript. We clarified this throughout the manuscript and also modified the model Figure accordingly (new Fig. 5f). We now write (page 18, line 455): “*Our in vitro replication results (Fig. 4c, d and e) suggest that replication initiation, rather than elongation, is affected by impaired DDK-dependent phosphorylation of Arp8. Further work is needed to reveal which subpopulation of origins respond to this mechanism in our purified, reconstituted system. It will be interesting to determine the extent to which nucleosome positioning contributes to the firing of an origin early, and the extent to which other chromatin features, such as histone modifications or 3D organisation, play a role. While this remains to be proven, we cautiously speculate that the replication timing programme is fine-tuned not only at the level of competition between replication origins for the early or late recruitment of replication factors, but also at the level of the early or late establishment of nucleosome organisation at origins conducive to the initiation of replication.*”; and (page 21, line 513): “*We propose that the accurate organisation of nucleosomes is a key factor in replication initiation and may therefore contribute to the replication timing programme.*”

We hope the reviewer will concur that this has enhanced the clarity of the manuscript and will maintain readers' enthusiasm for future developments in this area.

6. Discussion page 17, lines 419-421 : missing word ? I do not fully agree on that since it is not clear if arp8-AA exhibit replication defects, G1/S transition defects or both ?

We hope that our new experiments and changes to the text will now clarify these issues.

7. Figure 5a legend: cell cycle analysis by PI staining (or other dye) is not FACS but flow-cytometry, please correct.

Corrected.

8. Resolution of Extended data figure 2 is poor on my PDF version, please adjust for the final version.

We agree and changed the whole Figure (new Extended data Figure 2).

We would like to sincerely thank all three reviewers for taking the time to review our manuscript. We are pleased that Reviewer Nos. 2 and 3 consider the paper ready for publication. We also acknowledge Reviewer No. 1's concerns; however, we respectfully note that the points raised have already been addressed in the revised manuscript.

Reviewer #1 (Remarks to the Author):

The revised manuscript fails to address the core issue regarding the direct role of DDK phosphorylation on INO80 activity. The authors' conclusions rely entirely on comparing the INO80-AA mutant to the wild-type complex. This approach can only demonstrate that Ser65 and Ser233 of Arp8 are important for INO80's structural integrity and function; it cannot establish a direct regulatory role for DDK phosphorylation of Arp8.

To substantiate their claim, it is essential to perform their key assays with INO80 in the presence and absence of active DDK related to Figures 2 and 3. Only a direct comparison between INO80 and DDK-phosphorylated INO80 can determine whether DDK phosphorylation directly modulates INO80's ATPase and nucleosome remodeling activities.

It seems that this reviewer did not realize the data in our manuscript that is relevant to his/her concern. Reviewer 1 writes "*The authors' conclusions rely entirely on comparing the INO80-AA mutant to the wild-type complex.*", and suggests "*To substantiate their claim, it is essential to perform their key assays with INO80 in the presence and absence of active DDK related to Figures 2 and 3.*" However, we note that we already did exactly this.

The key and most physiologically relevant assay, which monitors chromatin remodeling as a combination of ATPase activity, sliding and regular spacing of nucleosomes, is presented in Figure 4. We assemble nucleosome arrays on a library of all yeast origins and measure the activities of wild-type INO80 and the INO80-AA mutant complexes in generating phased arrays of regularly spaced nucleosomes. We agree with Reviewer 1 that this comparison, while strongly supportive of a role of DDK-phosphorylation, may also, at least in part, reflect some effects due to the two amino acid substitutions in the INO80-AA mutant complex. Therefore, exactly as suggested by Reviewer 1, we confirmed a direct role of DDK-dependent phosphorylation by comparing only wild-type INO80 complexes, but in their unphosphorylated, i.e. phosphatase-treated, versus phosphorylated, i.e. DDK-treated, forms in this key assay (Supplementary Figure 6). This is exactly the type of experiment referenced by the reviewer. These experiments clearly demonstrated that the effects on nucleosome remodeling were directly due to phosphorylation.

Further, this assay is the key assay (as required by Reviewer 1) as it is most relevant for INO80 function in the physiological context. Other assays, such as those on mononucleosomes, are more remote from the physiological situation and were included by us to allow comparisons with previously published work and support the overall narrative. Our central conclusion is most strongly supported by the experiments in Figure 4 and Supplementary Figure 6, which were conducted using both strategies: with the INO80-AA mutant complex and with the in vitro DDK-phosphorylated wild-type INO80 complex. We do not see a need to repeat this two-pronged approach for all other assays, as the assay most critical for physiological function already supports the main conclusion.

For these reasons, we are convinced that the current version of the manuscript comprehensively addresses the concerns of all three Reviewers.

However, we have added one sentence to acknowledge the formally possible, though in our view unlikely, involvement of another kinase. We now state (page 14): “*Although all our in vivo and in vitro data strongly suggest that DDK targets INO80, we cannot formally exclude the possibility that another kinase could be involved as well.*”

Reviewer #2 (Remarks to the Author):

The authors have address all my comments completely. This paper is now ready for publication.

Many thanks!

Reviewer #3 (Remarks to the Author):

In response to my comments, the authors provided new experimental results that reinforce the message of the manuscript. I am recommending the manuscript for publication without hesitation.

The only information missing from the manuscript is a list of the genes used in the RNA-seq analyses for 'cell cycle associated genes' and 'DNA replication associated genes'. It is important that readers have access to this information.

Many thanks! We have provided a comprehensive list of all cell cycle and replication genes with the submission, as requested.